# Acetylation of histones and non-histone proteins is not a mere consequence of ongoing transcription

Tim Liebner [1], Sinan Kilic [1], Jonas Walter [1], Hitoshi Aibara[1], Takeo Narita [1] & Chunaram Choudhary [1]✉

In all eukaryotes, acetylation of histone lysine residues correlates with transcription activation. Whether histone acetylation is a cause or consequence of transcription is debated. One model suggests that transcription promotes the recruitment and/or activation of acetyltransferases, and histone acetylation occurs as a consequence of ongoing transcription. However, the extent to which transcription shapes the global protein acetylation landscapes is not known. Here, we show that global protein acetylation remains virtually unaltered after acute transcription inhibition. Transcription inhibition ablates the co-transcriptionally occurring ubiquitylation of H2BK120 but does not reduce histone acetylation. The combined inhibition of transcription and CBP/p300 further demonstrates that acetyltransferases remain active and continue to acetylate histones independently of transcription. Together, these results show that histone acetylation is not a mere consequence of transcription; acetyltransferase recruitment and activation are uncoupled from the act of transcription, and histone and non-histone protein acetylation are sustained in the absence of ongoing transcription.

Acetylation of the ε-amino group of lysine (hereafter acetylation) is a major posttranslational modification (PTM) that was first discovered in histones[1]. Proteome-wide analyses reveal that acetylation is a frequently occurring PTM that targets thousands of sites in mammalian cells[2–5]. Protein acetylation is implicated in the regulation of diverse cellular processes[6–9], but the function of acetylation is most extensively studied in the context of histones and gene regulation. Over the past six decades, tens of thousands of studies have investigated the link between histone acetylation and gene transcription[7,8,10–12]. Early studies showed that acetylation of histones is insensitive to transcription inhibition[13,14], that an increase in histone acetylation coincides with signal-induced gene activation[15–17], that the activity of acetyltransferases is required for gene activation and cell viability[18–21], and that inhibition of the p300/CBP acetyltransferase activity by the viral protein E1 inhibits p300-dependent transcription in vivo[22]. These studies indicated a close connection between acetylation and transcription. However, in these historical analyses, the lack of knowledge about the site- and locus-specificity of acetylation and poor temporal resolution of genetic analyses made it difficult to discern whether histone acetylation occurred before or after transcription activation.

Enabled by the advances in next-generation sequencing technologies, site-resolved, genome-wide analyses showed that histone acetylation is prominently enriched in active promoters and enhancers and that the level of promoter acetylation correlates with the level of transcription[23–25]. Histone acetylation appears to precede gene activation[26,27]. Notably, acute inhibition of CBP/p300 acetyltransferase activity inhibits the transcription of hundreds of genes within minutes, and the transcription inhibition is rapidly reversed by subsequent inhibition of deacetylases[28]. The strong correlation between histone acetylation and transcription supports the model that histone

[1]Department of Proteomics, The Novo Nordisk Foundation Center for Protein Research, Faculty of Health and Medical Sciences, University of Copenhagen, Blegdamsvej 3B, 2200 Copenhagen, Denmark. ✉e-mail: chuna.choudhary@cpr.ku.dk

acetylation precedes gene activation and may play a regulatory role in transcription[8,10–12,29].

An alternative model posits that histone acetylation occurs as a consequence of ongoing transcription[30]. This model draws support from studies showing that active transcription promotes the recruitment and/or activation of acetyltransferases. For example, transcription-induced formation of R loops facilitates the binding of acetyltransferase TIP60[31]. Also, transcription promotes TIP60 recruitment through interaction with phosphorylated RNA polymerase II (RNAPII)[32]. The acetyltransferase CBP/p300 is activated by enhancer RNAs (eRNAs), which are very short-lived, implying a role of ongoing transcription in CBP/p300 activation[33]. Possibly the strongest and most direct evidence supporting this model comes from the recent work showing that acute inhibition of transcription causes a global loss in histone acetylation[34,35]. This has led to the suggestion that RNAPII activity directs the deposition and maintenance of histone modifications associated with active transcription and that histone acetylation reflects transcription activity rather than posing a locus for future activity or directly regulating gene transcription[36]. In this model, histone acetylation primarily occurs as a consequence of transcription, and ongoing transcription is required to sustain histone acetylation.

The model that ongoing transcription is required for the recruitment and/or activation of acetyltransferases and histone acetylation raises several fundamental questions. (1) Is transcription required for the acetylation of all histone sites or specific sites? (2) Does transcription specifically activate acetyltransferases for catalyzing histone acetylation? (3) What is the impact of transcription on the acetylation of non-histone proteins? The last question is pertinent because the same acetyltransferases, such as GCN5/PCAF and CBP/p300, can catalyze acetylation on histone and non-histone proteins[6,37–39]. Given the broad interest in acetylation and transcription, addressing these questions is crucial for delineating the intricate relationship between histone modifications and gene regulation.

Here we use quantitative mass spectrometry and three distinct small-molecule inhibitors of transcription to quantify the proteome-wide relationship between transcription and protein acetylation. We find that the global acetylation patterns remain unchanged by transcription inhibition. In transcriptionally inhibited cells, simultaneous inhibition of CBP/p300 strongly reduces the acetylation of the target histone sites, conclusively demonstrating that acetyltransferases remain active in the absence of ongoing transcription. Our results challenge the view that histone acetylation is mainly a consequence of ongoing transcription.

## Results

### Proteome-scale quantification of lysine acetylation

We used SILAC (stable isotope labeling by amino acids in cell culture)-based mass spectrometry to quantify changes in lysine acetylation after acute transcription inhibition. SILAC-labeled mouse embryonic stem cells (mESCs) were treated with either vehicle control (DMSO), actinomycin D (ActD), or NVP-2 for 2 h. These inhibitors impact transcription by distinct mechanisms: ActD binds to DNA and prevents RNAPII binding and elongation[40], while NVP-2 inhibits the CDK9 kinase catalytic activity and prevents RNAPII pause release[41].

The cells were treated with the inhibitors for 2 h, protein lysates from the inhibitor and control-treated cells were mixed in equal amounts, proteolyzed with trypsin, acetylated peptides were enriched using pan-anti-acetyl-lysine antibodies and analyzed by liquid chromatography-coupled tandem mass spectrometry (LC-MS/MS) (Fig. 1a). We performed six independent biological replicates, and cumulatively quantified a total of 20,942 sites (Fig. 1b), presenting one of the deepest acetylome datasets in a single cell type. More than half of the sites were quantified in at least three biological replicates, and over three-quarters of sites were quantified in at least 2 biological replicates (Fig. 1c). For further analyses, we included sites that were identified in at least three biological replicates (ActD, $n = 11,916$; NVP-2, $n = 11,910$).

To ensure accurate quantification, we separately quantified SILAC ratios of proteins and used the median protein ratios to normalize acetylation site ratios. We used a cut-off of >2-fold change in acetylation to define regulated sites. Surprisingly, less than 0.25% of the quantified sites showed >2-fold change in acetylation (Fig. 2a). The small fraction of regulated sites is within the normal range of quantitative error in SILAC experiments. Supporting this notion, the fraction of sites showing a decrease and increase in acetylation is very similar. Sites directly acetylated due to transcription would be expected to be similarly affected by different transcription inhibitors. Among the sites quantified across both inhibitors, just three sites were downregulated, and two sites were upregulated (Fig. 2b). This suggests that ongoing transcription has a negligible impact on global protein acetylation.

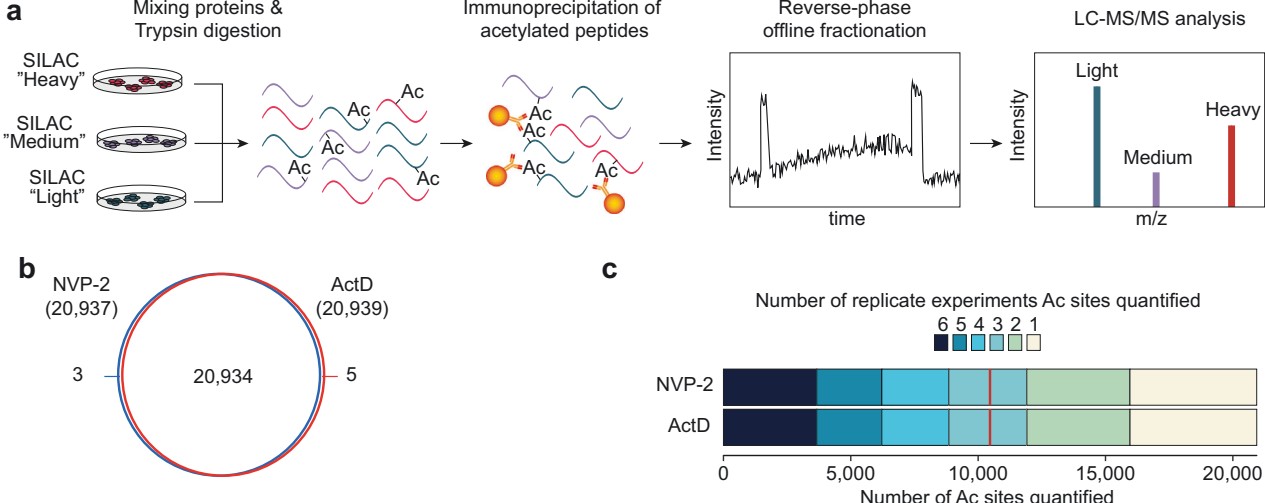

**Fig. 1 | The strategy and summary of global acetylome analysis after acute transcription inhibition. a** SILAC based proteomics workflow to quantify lysine acetylation sites. **b** Venn diagram showing the overlap between acetylation sites quantified in after treatment with ActD (1 µg/mL) and NVP-2 (100 nM). Cells were treated with the indicated inhibitors for 2 h and acetylation changes quantified as shown in schematic (**a**). **c** Shown is the number of biological replicates in which acetylation sites were quantified. A total of 6 biological replicates were performed.

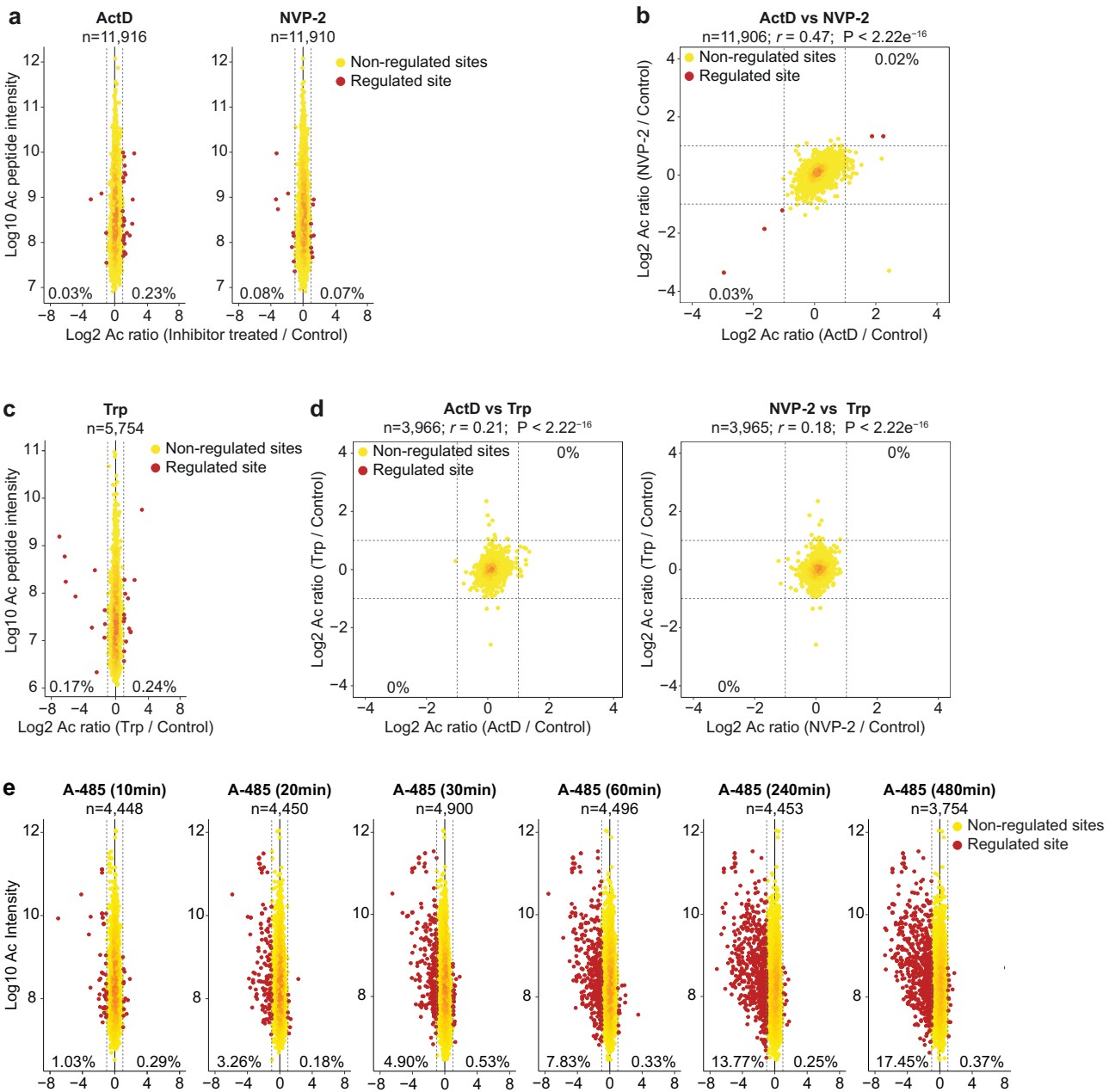

**Fig. 2 | Acetylation is not impaired by acute transcription inhibition but reduced after CBP/p300 inhibition. a** Shown is the change in acetylation site abundance after the treatment ActD and NVP-2 and MS intensity of the acetylated peptides. The cells were treated with ActD (1 μg/mL) and NVP-2 (100 nM) for 2 h, and change in acetylation was quantified by SILAC-based MS. The number (*n*) of acetylation sites quantified and the fraction of sites showing ≥ 2-fold down- or up-regulation is indicated. The dotted line indicates ≥ 2-fold up- or downregulation. **b** Correlation between acetylation site changes after treatment with ActD and NVP-2. The dotted line indicates ≥ 2-fold up- or downregulation. The number (*n*) of overlapping acetylation sites quantified in both conditions, Pearson's correlation (*r*), the corresponding *p*-value (*P*) from a two-sided test and the fraction of sites showing ≥ 2-fold down- or up-regulation in both conditions is indicated. **c** Shown is

the change in acetylation site abundance after the treatment with Trp (500 nM) and MS intensity of the acetylated peptides. **d** Correlation between acetylation site changes after treatment with ActD and Trp (left panel) and NVP-2 and Trp (right panel). The dotted line indicates ≥ 2-fold up- or downregulation. The number (*n*) of overlapping acetylation sites quantified in both conditions, Pearson's correlation (*r*), the corresponding *p*-value (*P*) from a two-sided test and the fraction of sites showing ≥ 2-fold down- or up-regulation in the conditions is indicated. **e** Shown is the change in acetylation site abundance in mouse embryonic fibroblasts treated with CBP/p300 inhibitor A-485 for the indicated time. The number (*n*) of acetylation sites quantified and the fraction of sites showing ≥ 2-fold down- or up-regulation is indicated. The shown data are re-analyzed from Weinert et al.[40].

To further confirm this result, we used an additional inhibitor to abrogate transcription. We chose triptolide (Trp) because it binds to a molecular target different from ActD and NVP-2. Trp inhibits TFIIH and prevents transcription initiation[42]. Nascent transcription analyses confirmed robust inhibition of global transcription by triptolide (Supplementary Fig. 1, Supplementary Data 1). We used the above-described SILAC-based approach and performed two replicate acetylome experiments with Trp. Despite globally downregulating transcription, the impact of Trp on acetylation was negligible, with just a few sites showing down- or up-regulation after 2 h of Trp treatment (Fig. 2c, Supplementary Data 1). If acetylation were to be truly regulated by transcription, then we would anticipate that the regulated

sites would show consistent change after transcription inhibition by different inhibitors. However, no sites were found that were consistently downregulated by Trp, ActD, and NVP-2 (Fig. 2d). These results show that the global protein acetylation landscapes remain unchanged after acute transcription inhibition. This contrasts sharply with previously observed acetylation changes after acute inhibition of the acetyltransferase CBP/p300 by A-485[39]. In contrast to downregulation of <0.25% sites after 2 h of ActD and NVP-2 treatment (Fig. 2a), CBP/p300 inhibition decreased acetylation of ~8% sites within 1 h of inhibitor treatment (Fig. 2e). Site-specific, and time-dependent decrease in histone site acetylation by CBP/p300 inhibition demonstrates that our mass spectrometry approach is suitable for detecting changes in histone acetylation. In the same expetimental setup, histone acetylation is not decreased after transcription inhibition, showing that the lack of regulation is not due to technical limitations.

### Histone acetylation is uncoupled from ongoing transcription

Because of the intense interest in histone acetylation and given the reports of the strong coupling between ongoing transcription and histone acetylation[34,35], we took a deeper look into the acetylation of histones. We quantified over three dozen acetylation sites in core-histones that were measured in replicate measurements (2 replicates for Trp, and 6 replicates for ActD and NVP-2) (Fig. 3a). Consistent with the lack of major global changes in acetylation, the acetylation of core-histones remained almost unaltered after transcription inhibition. This sharply contrasts with the strong downregulation of histone sites after CBP/p300 inhibition (Fig. 3b).

### CBP/p300 acetylate histones in the absence of transcription

We used quantitative image-based cytometry to complement and independently verify the results obtained by mass spectrometry. We analyzed the acetylation of H3 K27 (H3K27ac), which marks both promoters and enhancers and has previously been reported to be dependent on ongoing transcription[34,35]. To demonstrate the sensitivity and accuracy of our measurements, we included two important controls in the analyses. Firstly, we used A-485 to acutely inhibit the activity of CBP/p300[43]. A-485 treatment strongly impairs H3K27ac[26,39]. If the acetylation of these sites was solely a consequence of transcription, inhibition of transcription and CBP/p300 would be expected to cause a similar reduction in H3K27ac. Secondly, we analyzed histone H2B K120 ubiquitylation (H2BK120ub). H2BK120ub is deposited co-transcriptionally by the heterodimeric ubiquitin ligase RNF20/40[44]. It has been shown that inhibition of transcription by ActD and 5,6-dichloro-1-beta-D-ribofuranosylbenzimidazole (DRB) strongly reduces H2BK120ub[45]. Thus, H2BK120ub serves as a valuable reference for confirming the effective inhibition of transcription by the used inhibitors and for analyzing the transcription dependency of other histone marks. Unlike global transcription inhibitors, A-485 impairs the transcription of only a subset of the actively transcribed genes[28], and thus, A-485 is not expected to cause a strong global reduction in H2BK120ub.

As expected, inhibition of transcription by ActD, Trp, and NVP-2 nearly abolished H2BK120ub (Fig. 4a, b), confirming the efficacy of the inhibitors and the suitability of the assay for detecting changes in histone modifications. In contrast to H2BK120ub, ActD, Trp, and NVP-2 treatments did not cause any measurable decrease in H3K27ac (Fig. 4a, c). On the other hand, CBP/p300 inhibition by A-485 led to a strong reduction in H3K27ac, but not in H2BK120ub (Fig. 4a–c). H3K27ac was similarly impacted in cells treated with A-485 alone and cells treated with the combination of A-485 and one of the transcription inhibitors, but H2BK120ub was reduced only in the combined treatments (Fig. 4a–c). This result was confirmed using an independent H3K27ac antibody clone (Supplementary Figs. 2, 3, Supplementary Data 1). These results and the global acetylome analyses unambiguously demonstrate that acetyltransferases remain active and continue to acetylate histones in the absence of ongoing transcription.

### Genome-wide H3K27ac is not reduced by transcription inhibition

Two prior studies showed that transcription inhibition reduces the acetylation of histones in bulk[34,35]. Acetylation of H3K27ac was strongly reduced globally in ChIP-seq without a decrease in nucleosome occupancy in enhancer regions[35], which exhibit high levels of H3K27ac and H2B acetylation enrichment[46,47]. To test if H3K27ac is regulated in specific genomic regions, we analyzed H3K27ac enrichment by ChIP-seq in control cells as well as in cells treated (2 h) with ActD and NVP-2. H3K27ac was detected in promoters of most of the actively transcribed genes, but detected in only a small fraction of genes that are not expressed or expressed below the detection threshold of our assay (Fig. 5a). We found no appreciable and consistent decrease in H3K27ac after transcription inhibition by NVP-2 or ActD (Fig. 5b, c, Supplementary Fig. 4, Supplementary Data 1, Source Data), regardless of whether we normalized data using a spike-in reference or the rpkm-based approach. If anything, NVP-2-treated cells exhibited a very small increase in H3K27ac in active promoters.

In contrast to the lack of H3K27ac downregulation by transcription inhibition, using the same approach, we previously found that H3K27ac and H2B N-terminus acetylation sites are rapidly (<15 min) reduced after inhibition of CBP/p300 activity[47]. Furthermore, transcription inhibition by NVP-2 specifically increased H2BK20ac in actively transcribed gene body regions[47]. Because H2BK20ac in these analyses was increased after the treatment with NVP-2 treatment, it further supports that CBP/p300 remains active in transcription inhibited cells.

The transcription inhibition-induced reduction in H3K27ac has been rationalized by reduced acetyltransferase function[34,35]. CBP/p300 chromatin recruitment and activation is regulated by its binding to transcription factors[48]. CBP/p300 is autoacetylated within its activation loop[49], and catalytic inhibition of CBP/p300 rapidly reduces acetylation within its activation loop region[39]. In contrast, transcription inhibition did not cause an appreciable decrease in acetylation sites in CBP/p300 (Supplementary Data 1). This further indicates that CBP/p300 remains active in after transcription inhibition, and this may explain why we do not observe a decrease in H3K27ac in our assays.

## Discussion

The work presents proteome-scale quantification of lysine acetylation after acute transcription inhibition, revealing that histone and non-histone protein acetylation remains largely unchanged after transcription inhibition.

Our findings are consistent with early studies showing that histone acetylation is not affected by transcription inhibition[13,14,50,51]. However, our results contrast with some of the recent reports showing that histone acetylation is strongly reduced after transcription inhibition[34,35]. The reason for this discrepancy is unclear. One of the studies showed that H3K27ac and H3K27me3 were similarly strongly decreased within 1 h of transcription inhibition[35]. This is perplexing, given that the half-lives of H3K27ac and H3K27me3 are very different in mammalian cells[52,53]. The rapid reduction of H3K27me3 in EZH2-bound peak regions was rationalized by transcription-dependent recruitment of EZH2[35,54]. However, H3K27me3 decreases very slowly after the chemical or genetic ablation of EZH2 function, and the rate of H3K27me3 reduction roughly equals the time of the cell division cycle, indicating that the decrease in H3K27me3 after EZH2 activity ablation occurs by replication-coupled histone dilution rather than active demethylation[55,56].

While ongoing transcription is dispensable for global protein acetylation, we acknowledge that active transcription can and does

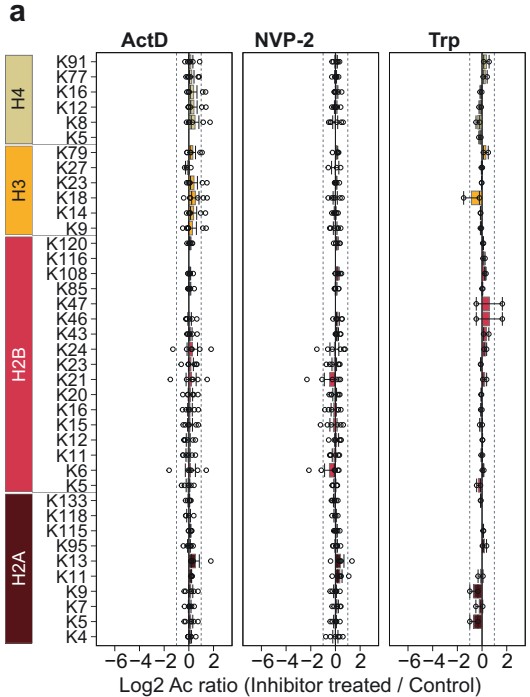

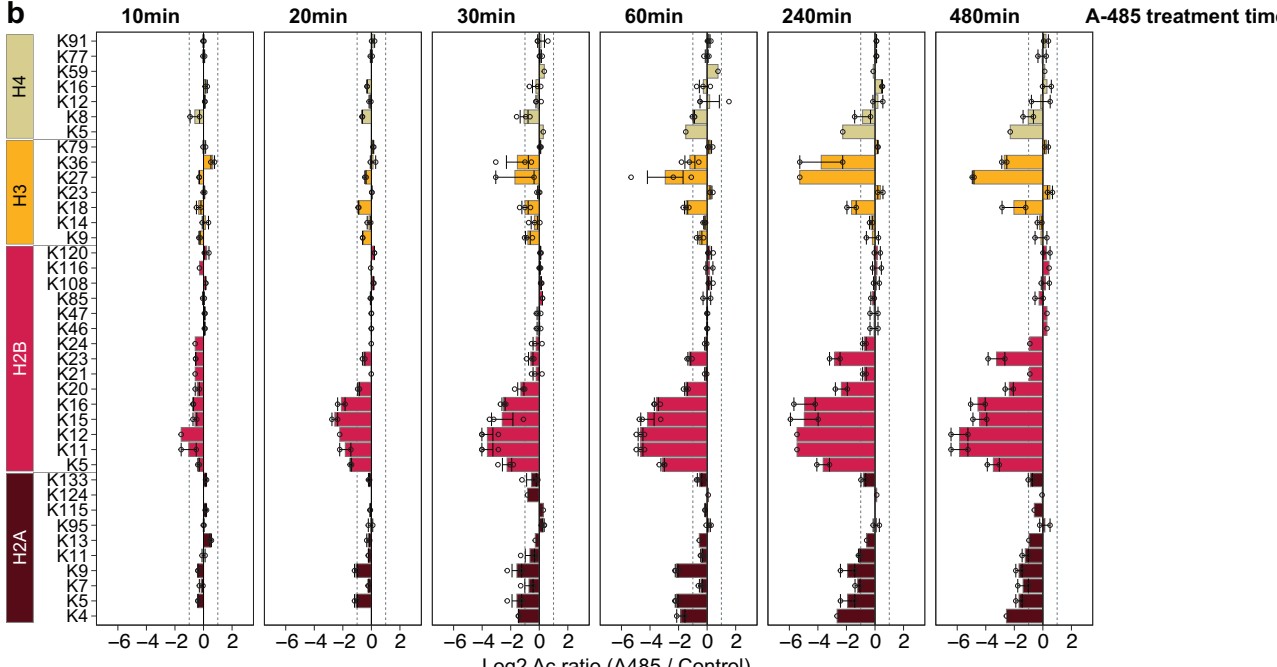

**Fig. 3 | Acetylation of core histones is site-selectively reduced after CBP/p300 inhibition, but remains unaltered after acute transcription inhibition.**
**a**, **b** Acetylation sites on core-histones quantified in this study (**a**), and in A485-treated MEF cells from Weinert et al.[40]. (**b**). Bar plots show distribution of the data. Bars show the mean log$_2$ acetylation site ratio of sites occurring in the indicated core-histones. Data points for each replicate are shown and error bars indicate the standard error of the mean. For ActD and NVP-2 experiments six biological replicates ($n = 6$) were used. The study of Weinert et al.[40] used three biological replicates ($n = 3$).

affect histone modification patterns. H2BK120ub is deposited co-transcriptionally, and H2BK120ub is rapidly reduced after transcription inhibition. RNAPII elongation causes nucleosome remodeling, whereby one of the H2A-H2B dimers is exchanged, while the hexasome, containing one of the H2A-H2B dimer and both the H3-H4 dimers, is redeposited[57–59]. This leads to the loss of one of the H2A-H2B dimers. Consistently, we recently found that the acetylation of H2B sites is lower in actively transcribed regions than H3K27ac, and

transcription inhibition increases the acetylation of histone H2B in these regions[60].

We further recognize that the effects of short- and long-term transcription inhibition will be different. Longer-term inhibition of transcription will inevitably impact protein expression and, consequently, will affect histone acetylation. For example, acetylation of H3K27ac remained stable after 4 h of triptolide treatment[50], but acetylation of several histone H3 sites was reduced in cells treated with

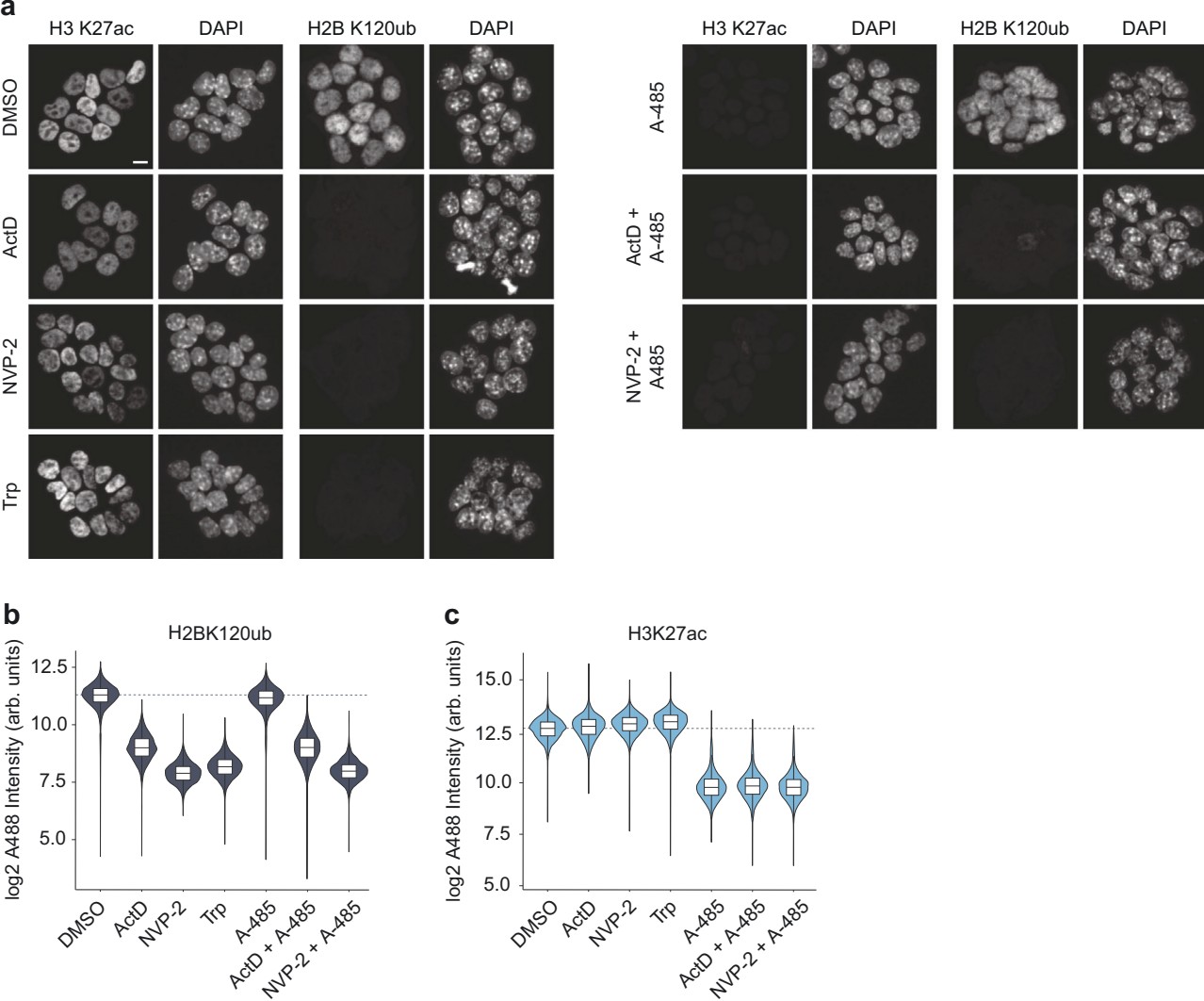

**Fig. 4 | Transcription inhibition and CBP/p300 inhibition differently impair H2BK120ub and histone acetylation. a** Representative micrographs of mouse embryonic stem cells treated with CBP/p300 inhibitor A-485 or the indicated transcription inhibitors (concentrations: A-485 10 μM, ActD 1 μg/mL, Trp 5 μM, NVP-2 1 μM). Cells were treated with the indicated inhibitors for 2 h, stained with the indicated histone marks, and analyzed by immunofluorescence. Scale bar is 10 μm. **b, c** Quantification of change in H2BK120ub (**b**) and H3K27ac (**c**) after the indicated treatments, in 6000 automatically selected cells per condition and staining. Median nuclear staining intensity relative to the DMSO control for the different combinations of treatments and staining as determined from image-based cytometry analysis of 6000 cells per combination. The dotted line indicates the median intensity of the DMSO treated cells. Violin plots show the distribution of the data, and the box represent the 25th and 75th percentiles as lower and upper hinges, with the bar within box indicates median. The whiskers show 1.5 × IQR.

DRB for 12 h[61]. Indeed, many transcription factors, such as MYC, p53, and HIF1A, are relatively short-lived, and ongoing transcription would be expected to be required to sustain their transcription. Decreased expression of transcription factors could impair the recruitment and/or activation of acetyltransferases. For this reason, it is impossible to completely rule out a role of transcription in protein acetylation. Because transcription inhibition does not cause a noticeable reduction in protein acetylation for up to 2 h, we suggest that the role of ongoing transcription in acetylation is minor, and the changes we measured are within the quantitative error of our measurements. We further show that CBP/p300 continues to acetylate H3K27ac in transcription inhibited cells, indicating that the recruitment and activation of these acetyltransferases are uncoupled from transcription. Since most acetylation appears to occur independently of ongoing transcription, histone acetylation should not merely be considered a consequence of ongoing transcription.

We want to clarify that our approach involves the use of trypsin protease, and our chromatography setup will likely not detect peptides that are either too short or too long (<5, or over ~30 amino acids). Therefore, our analyses are not comprehensive, and there may be sites, or specific combinations of sites, in histones and non-histone proteins that may be regulated by transcription inhibition but are not quantified in our study. Notwithstanding this generic limitation, our analyses provide one of the deepest coverages of global acetylation and include a great majority of known sites in histones. For the sites we could quantify, we did not find a measurable decrease by transcription inhibitors.

Additionally, our analysis of protein acetylation, including histones, is conducted in bulk. It is hypothetically possible that transcription inhibition may decrease acetylation at specific histone sites or combinations only or that the loss of acetylation in some genomic regions is compensated by an increase in acetylation in others.

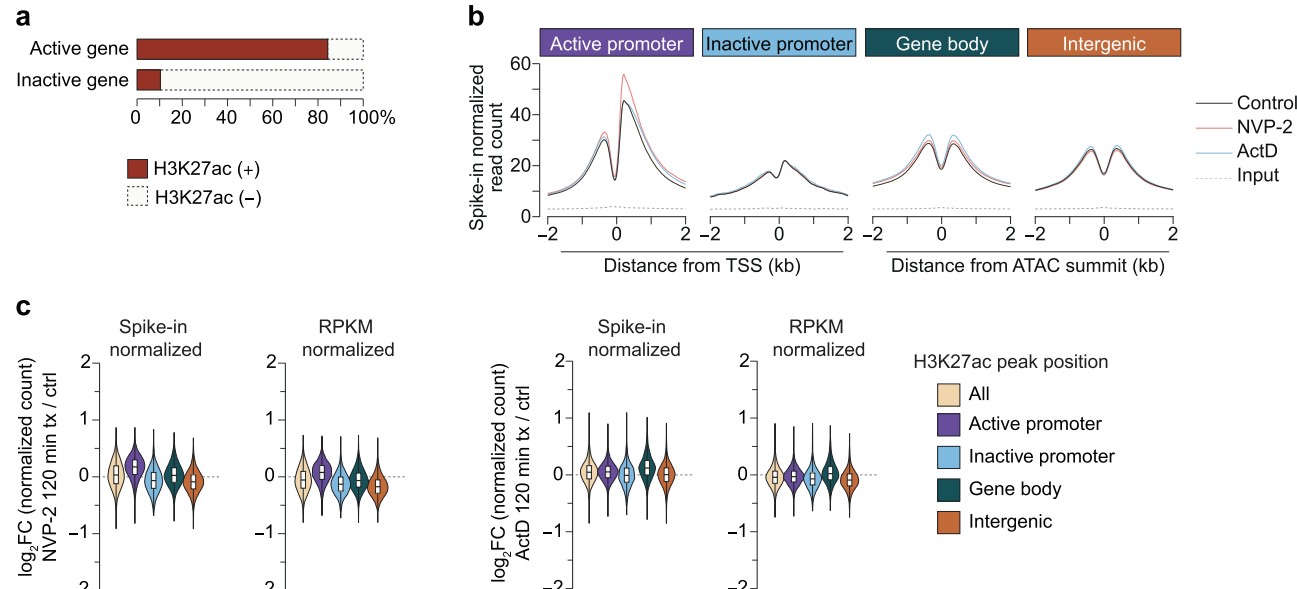

**Fig. 5 | Acute transcription inhibition leaves genes carrying H3K27ac mark unaltered. a** Percentage of active and inactive genes promoters marked by H3K27ac. Active promoters (±1 kb from the TSS) are defined as those mapping to genes expressed in mESCs (5-ethynyluridine (EU) RNA sequencing (RNA-seq) transcripts per million (TPM) > 2) and marked with H3K4me3, while inactive promoters are defined as promoters that do not meet the criteria for active promoters. **b** Shown are average profiles of H3K27ac spike-in normalized read counts in control and transcription inhibited mESCs at different genomic regions. Cells were treated for 2 h (ActD 1 μg/mL or NVP-2 1 μM). Active and inactive promoters are defined as described in Fig. 5a. Other genomic regions were classified as the following categories: all, all peaks; gene body, peaks occurring within gene bodies; gene bodies; intergenic, peaks occurring outside promoters and gene bodies. Promoter peaks are shown in dependence of the transcription-start-site (TSS). Gene body and intergenic peaks are shown relative to the proximal ATAC peak summit. **c** Relative fold change of H3K27ac site ChIP normalized read counts in mESC treated with NVP2 or ActD. Counts were either normalized to a spike-in or reads per kilobase million (RPKM). The dotted lines indicate a fold-change of 0. Violin plots show the distribution of the data, and the box represent the 25th and 75th percentiles as lower and upper hinges, with the bar within box indicates median. The whiskers show 1.5 × IQR. All the ChIP-seq experiments were performed with 2 biological replicates.

However, this explanation does not align with prior studies, which showed a global decrease in acetylation of all analyzed histone sites in transcription-inhibited cells, with no clear evidence for locus-specific regulation in ChIP-seq analyses[34,35]. Furthermore, different histone sites are acetylated by different acetyltransferases[62]. It seems unlikely that transcription inhibition would cause the relocation of all acetyltransferases to different chromatin regions, resulting in reduced acetylation in some regions and compensatory increased acetylation in others. Nonetheless, we could not rule this possibility out entirely. However, our H3K27ac ChIP-seq data found no evidence of redistribution of H3K27ac by transcription inhibition. Consistent with our data, another study found that genome-scale occupancy of H3K27ac is not reduced by triptolide (4 h) treatment[50]. Also, supporting independence between H3K27ac and ongoing transcription, it has been observed that acute depletion of MED14 broadly impairs transcription but causes little change in H3K27ac[51].

Some of the most frequently modified acetylation sites in mammalian cells are located in the N-termini of histones, and deacetylases regulate histone N-terminus sites more dynamically than sites located in the globular domain and C-terminus of histones[53,63–65]. An important question for future studies is to clarify the role of histone and non-histone acetylation in transcription. This work was not designed to evaluate the functional roles of acetylation in transcription, and we present no claim, explicit or implied, to suggest a functional role of acetylation in gene regulation. Because multiple histone and non-histone proteins are acetylated in vivo, deciphering the exact role of acetylation in transcription regulation is a daunting task that will require innovative approaches and community-wide efforts. Our results rule out a direct role of ongoing transcription in acetylation, narrowing down the number of hypotheses that need to be considered.

## Methods

### Cell culture

Mouse embryonic stem cells (mESC, Sigma Aldrich, ES-E14TG2a) were cultured in plates coated with gelatin (0.15% Bovine gelatin in PBS). Cells were grown in SILAC[66] Neurobasal media (custom-made, C.C.Pro GmbH, Oberdorla) supplemented with SILAC light, medium, or heavy labeled L-lysine HCL (84 mg/ml; Cambridge Isotope Laboratories Inc.) and L-arginine HCL (146 mg/ml; Cambridge Isotope Laboratories Inc.), as well as with 0.5 x N2 and 0.5 x B27 supplements (Thermo Fisher Scientific), 1,000 units/ml LIF (leukemia inhibitory factor, Merck Millipore), 1 μM PD0325901, 3 μM CT-99021 (custom-made by ABCR GmbH, Karlsruhe), 100 μM 2-mercaptoethanol, 150 μM sodium pyruvate. Supplemented SILAC amino acids had the following isotopic compositions: Light L-arginine-$^{12}C_6{}^1H_{14}{}^{14}N_4{}^{16}O_2$, Light L-lysine-$^{12}C_6{}^1H_{14}{}^{14}N_2{}^{16}O_2$, Medium L-arginine-$^{13}C_6{}^1H_{14}{}^{14}N_4{}^{16}O_2$, Medium L-lysine-$^{12}C_6{}^1H_{10}{}^2H_4{}^{14}N_2{}^{16}O_2$, Heavy L-arginine-$^{13}C_6{}^1H_{14}{}^{15}N_4{}^{16}O_2$, Heavy L-lysine-$^{13}C_6{}^1H_{14}{}^{15}N_2{}^{16}O_2$.

### Cell treatment, lysis, and peptide purification

SILAC-labeled mESCs were seeded onto a 245 mm square dish. After 2 days, the cells were treated either with DMSO (1:1,000; v/v) as a control, or the transcription inhibitors, 1 μg/ml ActD (Tocris Bioscience), 500 nM Trp (Sigma Aldrich), 100 nM NVP-2 (Tocris bioscience) for 2 h. For ActD and NVP-2, six biological replicates were performed (n = 6) and for TRP two biological replicates (n = 2). Cells were washed twice with PBS, followed by on-plate lysis with SDC lysis buffer (2% (w/v) sodium deoxycholate, 50 mM 4-(2-hydroxyethyl)-1-piperazineethanesulfonic acid (HEPES), pH 8.0). Cell lysates were boiled at 95 °C for 10 min, cooled down, and stored at -80 °C until further processing. Frozen cell lysates were then thawed and boiled again at 95 °C for 10 min and sonicated for 2 min

at 70% intensity (Branson Digital Sonifier SFX 150 1/8" Diameter Tapered µ tip). Cell lysates were centrifuged at 4,800 g for 10 min, the supernatant was transferred and the protein concentration was determined using BCA Protein Assay Kit (Thermo Fisher Scientific) according to the manufacturer's protocol. In one setup, lysates from differentially SILAC-labeled cells and treated with DMSO, NVP-2, and actinomycin D were mixed in equal amounts (based on protein content), and in another setup, lysates from SILAC-labeled cells treated with DMSO and triptolide were equally mixed. The mixed proteins were reduced by adding 1 mM DTT (dithiothreitol) and incubating at 95 °C for 15 min. Samples were cooled down and alkylated by adding 55 mM chloroacetamide (30 °C, 1 h). Subsequently, the samples were digested by adding trypsin 1:100 (w/w of protein) and incubating at 37 °C overnight, and sodium deoxycholate was removed by adding trifluoroacetic acid (TFA) to a concentration of 3% (v/v) and centrifuging at 4,800 g for 30 min. The peptides were purified on SepPack Classic C18 cartridges (Waters), according to the manufacturer's protocol. Peptides were eluted with 50% (v/v) acetonitrile (ACN). Acetonitrile was removed from the extracted peptides by vacuum concentrating at 60 °C until at least 50% of the initial volume was evaporated. A portion (~1 µg) of the peptide mix was taken aside for analyzing total protein expression, and the remaining peptides (~25 mg) were used for affinity purification of acetylated peptides.

## Pan-acetyl-lysine immune precipitation and fractionation

Acetylated peptides were affinity-enriched using agarose-conjugated pan-acetyl-lysine antibody beads (Cell Signalling Technology PTMScan Acetyl-Lysine Motif Kit). Before use, the beads were washed three times with 1 ml of 1xIAP NP-40 buffer (50 mM morpholinopropanesulfonic acid (MOPS) pH7.2, 10 mM sodium phosphate, 50 mM NaCl, 0.5% (v/v) NP-40). Peptides were reconstituted in 1xIAP NP-40 buffer, added to the beads slurry and incubated on a spinning wheel for 70 min. Subsequently, beads were centrifuged at 2,000 g for 1 min, and the supernatant was aspirated. The beads were washed twice with 1 ml IAP NP-40 buffer, twice with 1 ml IAP buffer (50 mM MOPS pH7.2, 10 mM sodium phosphate, 50 mM NaCl), and twice with 1 ml deionized water. Bound peptides were eluted by adding 100 µl of 0.15% (v/v) TFA and incubating at 30 °C for 5-10 min on a shaker (1,200 rpm). Beads were spun down at 2,000 g for 1 min, and the eluant was collected. The elution was repeated two more times. To remove any traces of detergents, the eluted peptides were cleaned using Stage-Tip-based SCX microcolumn[67]. The SCX micro-column was prepared by inserting 6 discs of SCX material (Empore Cation Solid Phase Extraction Disk) into a 200 µl tip. The Stage-Tip-based SCX micro-column was activated by washing with 100 µl of methanol. The micro-column was washed with 100 µl of SCX buffer (50% (v/v) ACN, 20 mM acetic acid, 20 mM boric acid, 20 mM phosphoric acid, pH9.0), and equilibrated with 100 µl of 50% (v/v) ACN in 0.1% (v/v) TFA. Peptides were loaded onto the micro-column and washed twice with 100 µl of 50% (v/v) ACN in 0.1% (v/v) TFA. Peptides were eluted from the micro-column using 100 µl of SCX buffer, vacuum-dried, and re-suspended in 20 µl 0.15% (v/v) TFA. Re-suspended peptides were fractionated with a C18 reverse-phase column (Phenomenex 2.6 µm EVO C18 100 Å, 150 mm×0.3 mm) on an EASY nLC 2000 (Thermo Fisher Scientific) coupled to an automated, custom-built fraction collector. Buffer A consisted of 1% (v/v) Triethylammonium bicarbonate (TEAB) in water and buffer B of 80% (v/v) ACN and 1% (v/v) TEAB in water. The sample was loaded onto the column at 2 µL/min at a maximum 500 bar, fractionated using the following gradient: 3% to 40% B in 57 min, 40% to 60% B in 5 min, 60% to 95% B in 10 min, 95% B for 10 min, 95% to 3% B in 3 min and finally back to 3% B for 8 min. Fractions were collected for 90 s which were concatenated into 12 fractions in total. The fractionated peptides were vacuum dried at 60 °C, and re-dissolved in 0.15% (v/v) TFA.

## Mass spectrometric analysis

After fractionation, peptides were analyzed by online liquid chromatography coupled with tandem mass spectrometry (LC-MS/MS) utilizing a Proxeon EASY-nLC1200 (Thermo Scientific) attached to an Exploris480 (Thermo Scientific). Peptides were loaded onto a chromatography column (length 15 cm, inner diameter 75 µm) packed with C18 reverse-phase material (Reprosil-Pur Basic C18, 1.9 µm, Dr. Maisch GmbH) and eluted by a 75 min gradient (5–40% (v/v) ACN/H$_2$O and 0.1% (v/v) formic acid). Eluting peptides were ionized by electrospray ionization and injected into the mass spectrometer under the following settings: 2.3 kV spray voltage, no sheath/aux/sweep gas flow rate, 270 °C capillary temperature and funnel RF at 40%. Xcalibur (v. 4.4.16.14) operated the Exploris480 and Orbitrap Exploris 480 Tune (v. 2.0.182.25) in data-dependent mode. MS1 scans were acquired at a mass resolving power of 120,000 (FWHM at 200 m/z) and a scan range of 300–1,750 m/z. The normalized AGC target for the scans was 300%, and the maximum injection time was 60 ms. Top 12 the most intense peptide ions were sequentially isolated and fragmented by higher-energy collisional dissociation (HCD) using a normalized collision energy of 25. MS2 scans were acquired with a mass resolving power of 15,000 (FWHM at 200 m/z), a normalized AGC target of 80%, a maximum injection time of 110 ms, and an isolation window of 1.3 m/z, and a fixed first mass of 100 m/z. The dynamic exclusion was set to 30 s, and only ions with a charge state between two and five were included in the analysis. Raw data were processed with MaxQuant (v. 2.1.0.0)[68] by a search against the *Mus musculus* UniProt database (downloaded May 3, 2021) for the SILAC pairs used in the experiments. The false discovery rate was kept at 1% for peptides and proteins. Cysteine carbamidomethylation was set as a fixed modification, and methionine oxidation, N-terminal acetylation, and lysine acetylation were set as variable modifications. Re-quantification was enabled and missed cleavages was set to two. Every other setting was unchanged to the default settings.

## Quantitative image-based cytometry cell preparation, acquisition, and analysis

Mouse embryonic stem cells (mESC, Sigma Aldrich, ES-E14TG2a) were seeded (12,000 cells/well in 200 µL media) in a 96-well plate (CellCarrier 96 Ultra, Perkin Elmer) coated with 50 µL Geltrex (1:100; v/v; in PBS) per well 10-20 min before seeding. After overnight attachment, cells were treated with DMSO (1:1000), ActD (1 µg/mL), A-485 (10 µM), ActD + -485, Trp (5 µM), NVP-2 (1 µM) and NVP-2 + A-485 for 2 h. Subsequently, cells were washed once with PBS and fixed for 20 min using 3.7% (v/v) formaldehyde for all conditions, except for staining with H2B K120ub where cells were fixed with methanol. After fixation, cells were washed with PBS and stored in the fridge overnight. Cells were permeabilized with 0.5% (v/v) triton X-100 for 5 min followed by blocking with antibody diluent (DMEM with 10% (v/v) fetal bovine serum and 0.02% (w/v) sodium azide, filtered) for 1 h. The permeabilized cells were incubated with the specified primary antibodies (1:1000, H3 K27ac: Rabbit monoclonal Anti-H3 K27ac, Abcam #ab177178; 1:1000, H3 K27ac: Rabbit monoclonal Anti-H3 K27ac (D5E4), Cell Signaling Technology, #8173; 1:1000, H2B K120ub: Rabbit monoclonal Anti-H2B K120ub, Cell Signaling Technology #5546) in antibody diluent for 2 h, washed thrice with PBS, incubated with secondary anti-rabbit A488 antibody (1:500; v/v) for 1 h, and finally stained with DAPI (0.5 µg/mL in PBS) for 10 min. Images were acquired on a Perkin Elmer Phenix spinning disk microscope equipped with two scMOS cameras (16 bit; 2,160 × 2,160 pixel; 6.5 µm pixel) using a 20x water objective (NA1.0 W Plan Apochromat/ WD 1.7 mm) and spinning disk confocal acquisition. For each well, 12 fields and 5 planes were acquired with the DAPI and Alexa 488 channels. Fixed settings were used to acquire all images across replicates.

## Image analysis

Images from each imaging channel, plane, and position were converted to hyperstacks and the planes were used to generate maximum

intensity projections using the Z project in Fiji ImageJ (1.53). 250 × 250 pixel examples from each condition were duplicated, the brightness/contrast set to the same across different treatments, and the resulting.tif files and corresponding.jpg files were saved.

Quantitative analysis (Supplementary Fig. 3) of the images was done with the Phenix Harmony analysis software. Acquired stacks were processed as maximum intensity projections and advanced flatfield and brightfield corrections applied. The input images from the A488 channel were background-corrected with the filter image module using the sliding parabola method with a curvature of 0.01. Nuclei were identified using the DAPI images using the C method in the software set with a common threshold of 0.40, area of 20 μm², splitting coefficient of 7.0, an individual threshold of 0.30, and contrast -0.50. Nuclei at the border of images were removed with the select population module using the common filter, remove border objects option. Identified nuclei and background-filtered A488 images were used to calculate the intensity properties of the antibody staining to determine the mean intensity per pixel in each nucleus.

## ChIP

Mouse embryonic stem cells (mESC, Sigma Aldrich, ES-E14TG2a) were treated 2 h with DMSO, 1 μg/ml Actinomycin D (Tocris Bioscience), or 100 nM NVP2 (Tocris Bioscience), respectively. After treatments, cells were cross-linked with 1% formaldehyde for 10 min, neutralized with 0.2 M glycine for 5 min, washed twice with PBS, then collected by cell lifter, spun down for 10 min at 3,000 rpm at 4 °C and stored at -80 °C until use. Cells were resuspended in SDS lysis buffer (10 mM Tris-HCl, pH 8.0, 150 mM NaCl, 1% (w/v) SDS, 1 mM EDTA, pH 8.0, cOmplete Protease Inhibitor Cocktail (Roche)) and sonicated by a Bioruptor Pico (30 cycles, 30 s on and 30 s off, Diagenode), then centrifuged for 10 min at 13,000×g at 20 °C for debris removal. The supernatants were stored at -80 °C. The aliquots of the supernatant were 5-fold diluted with ChIP dilution buffer (20 mM Tris-HCl, pH 8.0, 150 mM NaCl, 1 mM EDTA, 1% Triton X-100, cOmplete Protease Inhibitor Cocktail), incubated 20 h at 65 °C to reverse cross-link, then treated with 0.2 mg/ml RNase A (Thermo Fisher Scientific) for 30 min at 37 °C and proteinase K (Thermo Fisher Scientific) for 1 h at 55 °C. DNA fragments were isolated by phenol-chloroform extraction and recovered by ethanol precipitation with glycogen (Thermo Fisher Scientific). DNA fragments were quantified by Nanodrop (Thermo Fisher Scientific). The average size of DNA fragments was measured by 4200 TapeStation System (Agilent Technologies). For ChIP, the supernatants equivalent to 100 μg of mESC chromatin DNA, along with 10 μg of spike-in chromatin DNA from HEK293T, were incubated with 2 μg of anti-histone H3 acetyl K27 antibody (ab4729 Abcam) conjugated with Dynabeads M-280 sheep anti-Rabbit IgG (Thermo Fisher Scientific) by rotating overnight 4 °C. The immunoprecipitated Protein-DNA complexes were washed with ChIP dilution buffer, wash buffer high-salt (20 mM Tris-HCl, pH 8.0, 500 mM NaCl, 2 mM EDTA, 1% (v/v) Triton X-100, 0.1% (w/v) SDS), wash buffer low-salt (10 mM Tris-HCl, pH 8.0, 1 mM EDTA, 250 mM LiCl, 0.5% (w/v) sodium deoxycholate, 0.5% (v/v) NP-40) once respectively, washed with TE three times, then eluted with elution buffer (50 mM Tris-HCl, pH 8.0, 10 mM EDTA, 1% SDS). The immunoprecipitated DNA was reversed cross-linked and recovered from the eluates as described above.

## ChIP−seq library preparation

NGS library was prepared from 20 ng of ChIPed DNA measured by Qubit dsDNA HS Assay (Thermo Fisher Scientific) using the NEBNext Ultra II DNA Library Prep Kit for Illumina (New England Biolabs) according to the manufacturer's instructions. Adapter-ligated, U-excisioned DNA was cleaned by AMPure XP beads (Beckman Coulter) amplified by PCR 6 cycles, then cleaned again and size-selected (350–400 bp average size measured by Tapestation). 4 nM pooled Library DNA was sequenced on a NextSeq 2000 Sequencer (Illumina) as single-end 138 bp reads.

## Reference genome annotation

Mouse and human reference genome and annotation were downloaded from GENCODE (GRCm38 release 25)[37] and NCBI (T2T-CHM13v2.0), respectively. For reference, the transcript types of protein-coding and lincRNA were chosen as representative genes. Expressed gene were selected using the following criteria of Transcripts per million transcripts (TPM) > 2 in RNA-seq and the presence of H3K4me3 ChIP-seq peak within 1 kb from the transcription start sites (TSS). The longest isoform was selected if a single H3K4me3 peak is proximal to multiple transcripts of the same gene. Genes not meeting these criteria were defined as inactive genes.

## Processing RNA-seq data

Adapters were trimmed using Cutadapt[69]. Reads were aligned to the reference genome using STAR (version 2.6.1a)[70], excluding non-canonical junctions. Reads mapping to tRNA and rRNA regions were removed using Bedtools (version 2.23)[71]. The number of reads mapped to exons was quantified on a gene basis by using HTseq (version 0.11.1)[72]. TPM values were calculated using R.

## Processing EU-seq data

Adapters were trimmed using Cutadapt[69]. Bwa meme (version 1.0.4[73]), were used for reads mapping with soft clipping option for supplementary alignments. Low-mapping quality reads (MAPQ < 10) were removed using samtools (version 1.4)[74]. Reads mapped to rRNA and tRNA regions, obtained from UCSC genome browser, were excluded with Bedtools (version 2.23)[71]. The number of reads mapped to defined regions was counted using HTseq (version 0.11.1)[72]. Transcripts with low mapped reads (average reads ≤ 20 between replicates for both control and treatment conditions) were excluded from differential gene expression (DGE) analysis. Log2 fold-changes (FC) were calculated using the DESeq2[75]. Mitochondrial transcript reads were counted in 200 bp bins and regions where geometric mean counts among datasets > 20 were used for calculating scaling factors by normalizing the median of relative abundance.

## Processing of ATAC-seq data

Adapters were trimmed using Cutadapt[69]. Paired-end reads were aligned to the mm10 genome using BWA meme (version 1.0.4)[73] with soft clipping option for supplementary alignments. Duplicated read pairs were removed using Picard-tools (version 2.9.1, Picard Toolkit, 2019. Broad Institute, GitHub Repository. https://broadinstitute.github.io/picard/; Broad Institute). Low-quality reads (MAPQ < 10) and non-primary alignments were filtered out using Samtools (version 1.4)[74]. Peak regions were called using LanceOtron and its default model (wide-and-deep_jan-2021)[76]. Peak heights were calculated using deepTools2 bamCoverage function[77] with specified parameters (centerReads, bin size 20 bp, smooth length 400 bp, extend reads 200 bp, rpm normalization). Poorly- enriched peaks of peak summit height <8 rpm were filtered out.

## Processing of ChIP-seq data

Adapters were trimmed using Cutadapt[69]. Read sequences were aligned to the combined mouse mm10 and human T2T-CHM13v2 genome using bwa meme (version 1.0.4,)[73] with soft clipping option for supplementary alignments. Duplicated reads were annotated and removed using Picard toolkit tools (version 2.9.1, Picard Toolkit, 2019. Broad Institute, GitHub Repository. https://broadinstitute.github.io/picard/; Broad Institute). Low-mapping quality reads (MAPQ < 10) were excluded using samtools (version 1.4)[74]. Peak calling was conducted using LanceOtron with the default model (wide-and-deep_jan-2021)[76]. The peaks proximal within 2 kb are merged using Bedtools[71]. Peak

heights were calculated using deepTools2 bamCompare function[77] with the following parameters (centerReads, minMappingQuality 10, bin size 20 bp, smoothing length 400 bp, extension of reads to 200 bp, rpm normalization, and input rpm value is subtracted). The peak summit was defined as the center of the 20 bp bin at maximum height in each peak region. Poorly enriched peaks of maximum peak height at 20 bp bin <8 reads mapped per million (rpm) or read enrichment at +/- 500 bp around peak summit center <1 rpm after input subtraction were excluded. H3K27ac peaks that were not proximal to ATAC-seq peak summit within 400 bp, or whose peak summits were >1 kb from ATAC-seq peak summit were filtered out. The H3K27ac peak changes were quantified by analyzing the read count mapped to peak summit +/- 500 bp region. The FC ratios were calculated by using DESeq2 with incorporating scaling factors determined from the HEK293 cell spike-in reads counts. The aggregate ChIP-seq profiles were plotted using deepTools2 (version 3.5.2)[77]. For plotting the ChIP-seq signals, EnrichedHeatmap (Version 1.33.0)[78] was used.

## Statistical analysis

**Peptide and protein identification.** The Acetyl (K)Sites table was filtered to eliminate entries from reverse decoy databases and contaminant sources, as well as peptides with fewer than 2 SILAC ratio counts or a localization probability <0.9. Furthermore, the proteinGroups table obtained from the proteome measurements were additionally filtered to retain only proteins not identified by a modification site. Acetylation sites relative quantification was based on the SILAC Ratios extracted from the Acetyl (K)Sites table and normalized by the median SILAC ratio from the corresponding proteome measurement for each individual experiment to correct for mixing errors. For ActD and NVP2 experiments, we required that sites must be quantified in at least three biological replicates out of six, while for TRP experiments, a minimum of two out of two biological replicates were required. Scatter plots were generated by plotting the log10-transformed acetylated peptide intensity against the log2-transformed median of the treatment vs. control ratio. Additional scatter plots were generated by plotting the log2-transformed ratio of treatments vs. control for different perturbations against each other. To collapse acetylation site ratios for different histone isoforms, we combined ratios from different isoforms matching the site, and from each replicate experiment, we calculated their median ratio. Then we plotted median ratios from each replicate experiment, and displayed mean of the median ratios, and the standard error of mean. Sites showing ≥2-fold change (log2(1)), were considered as regulated. T-test was applied for sites that were quantified in at least three biological replicates and Benjamini-Hochberg method was applied for multiple correction comparison of the resulting p-values.

Two sided Pearson's correlation was calculated in R (version 4.3.1). R-generated violin, box, and bar plots show the distribution of the data. The boxplots represent the 25th and 75th percentiles as lower and upper hinges, with the median displayed as a bar within the box. The bars in the bar plots show the mean and the error bars show the standard error of the mean. The upper whisker extends from the upper hinge to the largest value within 1.5 times the interquartile range above it. The lower whisker extends from the lower hinge to the smallest value within 1.5 times the interquartile range below it.

**Sequencing analysis.** Unless otherwise mentioned, p-values are calculated using two-sided Mann–Whitney U-test and corrected for multiple comparisons using Benjamini & Hochberg method (R package stats version 3.6.2).

## Reporting summary

Further information on research design is available in the Nature Portfolio Reporting Summary linked to this article.

## Data availability

The mass spectrometry data have been deposited to the ProteomeXchange Consortium via the PRIDE[79] partner repository with the dataset identifier PXD044009. The publicly available data were used as followed: Mouse gene annotations from GENCODE. The following datasets from GSE146328 were reanalyzed: EU-seq in ESC control, ChIP-seq in ESC on H3K4me3 and Input control, and ATAC-seq in ESC. The sequencing raw data and genome browser tracks of EU-seq in Triptolide-treated ESC, and H3K27ac ChIP-seq data with or without NVP2- or actinomycin D-treated ESC generated in this study have been deposited in the NCBI Gene Expression Omnibus (GEO) database under accession code GSE260969. The imaging data generated in this study are provided in the Supplementary Data. Source data are provided with this paper.

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

## Acknowledgements

We thank the members of the Choudhary lab for their helpful discussions. We thank Elina Maskey for her excellent technical assistance. We thank the CPR Imaging Platform and the CPR Mass Spectrometry Platform for their assistance. This work was kindly supported by the following entities: the Novo Nordisk Foundation through the grant NNF22OC0074677 and NNF20OC0065482 to C.C., and the Lundbeck Foundation through the fellowship R347-2020-2170 to S.K. The Novo Nordisk Foundation Center for Protein Research is financially supported by the Novo Nordisk Foundation (no. NNF14CC0001).

## Author contributions

T.L. performed all mass spectrometry analyses and analyzed the data. S.K., J.W. and T.L. performed immunofluorescence analyses and analyzed the data. H.A. performed ChIP-analyses and T.N. analyzed the next-generation sequencing data. We thank Elina Maskey for her excellent assistance with performing H3K27ac ChIP, and Georgios Pappas for his help in the preparation of the next-generation sequencing library. With input from S.K., H.A, T.N, and J.W., T.L. and C.C. wrote the manuscript. C.C. conceived and supervised the project.

## Competing interests

The authors declare no competing interests.
