## [Peer Review File · Nature Communications]

Reviewers' Comments:

Reviewer #1:

Remarks to the Author:

In this manuscript Liebner et al aimed address the effect of transcriptional inhibition on histone acetylation levels by performing SILAC MS upon acute transcriptional inhibition.

Recent evidences show that inhibition of transcription leads to reduced level of histone acetylations at transcription sites and transcription shapes histone acetylation pattern at expressed genes.

Authors conclude that Acetylation on histones are not mere consequence of transcription and acetylation independent of transcription. Although authors findings are interesting and useful contribution to the field. However, measuring bulk acetylations using mass spec is not sufficient to conclude that acetylation is not mere consequence of transcription. I would encourage authors to perform additional experiments suggested below which will further strengthen their conclusions.

Major comments:

Assaying the bulk level of histone acetylations is not ideal to investigate the effect of transcription on histone acetylation at the transcription sites. What about the acetylation level at sites outside the transcribing genes, at regulatory elements. Can reduced histone acetylation at genes compensated by increased levels outside the genes?

Chromatin and histone acetylation level at regulatory elements and transcription start sites is highly dynamic, it is important to assay altered level of histone acetylations at different genomic features.

Controls missing to demonstrate that triptolide treatment is indeed inhibiting transcription within 2 hours and at the used concentration.

Can authors perform time course experiment by inhibiting transcription at different time point and assay the level of histone acetylations and association with transcription?

Authors lab has recently published their work on histone acetylations by performing ChIPseq for histone acetylation, since the lab has expertise and I expect authors to perform quantitative ChIPseq for H3K27ac and analyse the level at genes and intergenic sites.

Manuscript focus on H3K27ac, there are many acetylations on histone that are possibly altered upon perturbing transcription. For e.g. Like H3K36me3, H4K16ac is enriched on the gene bodies in the transcribed genes. It would be good to check if this marks is altered when transcription is inhibited.

minor comments: Abstract is too brief, please include organism and methodology used.

Reviewer #2:

Remarks to the Author:

Many thanks for the opportunity to review your manuscript. I enjoyed being able to take some time to read it and make myself at least superficially familiar with what seems to be an exciting and vibrant field. The introduction is very well presented and presents the apparently conflicting evidence in the sequentiality of histone acetylation and transcription clearly and comprehensively. More generally though, while on its own it is of interest to study the relationship between global protein acetylation and levels of transcriptional activity, it is not clear to me what the relevance is of acetylation of non-histone proteins to a discussion on causality of histone acetylation and transcription.

I wonder if the penultimate paragraph of the introduction would benefit from highlighting the possibility of recruitment of HATs to transcriptionally active loci, as mentioned earlier in the introduction, as a mechanism of histone acetylation as specific from global acetylation.

Methods & Results

I was specifically asked to comment on the imaging cytometry, and I will mainly restrict my specific comments to that part of the data as the SILAC mass spec is not an expertise of mine. I appreciate though from a brief look at the data that minimal, if any, changes were observed in global protein acetylation following inhibition of transcription, in contrast to convincing and time dependent changes following acetyltransferase inhibition. Figure 3 appears to present strong direct evidence that inhibition of transcription does not alter histone acetylation. Apologies if I missed this but why was the transcription inhibition done at a single time point of two hours but the CBP/p300 inhibition at multiple timepoints up to 1 hour?

The image acquisition and analysis are suited for the intended quantitative analysis, with specific reference made in the methods to the set image analysis parameters fixed for all images. The methods would also benefit from confirmation that all parameters set also fixed for image acquisition. Additionally it would be useful to see the associated DAPI images in parallel in figure 4 (as confirmation of the presence of cells) and an illustration of the masking and quantification strategy in supplementary data.

The presented data (Figure 4) shows a clear retention of signal detected with the H3K27ac antibody following exposure to transcriptional inhibitors.

Lines 291 and 292 refer to a supplementary table with details of the primary antibodies used. Apologies if I'm missing it but I can't find it. This is critical to assessing the quality of the IF analysis. I would also like to see experimental assessment of the specificity of the antibodies used for IF. Specificity of the H3K27ac antibody for H3K27ac against other acetylated proteins is essential to support the claim that CBP/p300 catalysed H3K27 acetylation is independent of transcription, to demonstrate the retained staining isn't due to non specific detection of other lysine acetylation.

I think the robustness of the conclusions from the IF needs to be toned down, especially in the context of the absence of data demonstrating the specificity of the H3K27ac antibody. It cannot be assumed based on the data currently presented that the Ab is detecting H3K27ac, or detecting H3K27ac alone. Line 143 to 156 I feel is not justified. While the IF results demonstrate convincingly that acetylated proteins are present in the presence or absence of transcription, they do not demonstrate that acetyltransferases continue to activate histones in the absence of ongoing transcription.

Reviewer #3:

Remarks to the Author:

This study by Choudhary and colleagues examined changes in histone acetylation levels after acute inhibition of transcription in mouse embryonic stem cells by mass-spec analysis. They used three different transcription inhibitors, which affected initiation, elongation, or RNA Pol II release. In contrast to previously published studies, some of which are referenced in the current studies, they found that transcription inhibition did not significantly reduce acetylation. As expected, inhibiting CBP/P300 activity resulted in significant reductions in acetylation at various sites, including the core histones. Based on these studies they conclude histone acetylation is uncoupled from the transcription. The authors have tried to address an important question—the link between histone acetylation and transcription. However, their approach and results are inadequate in supporting their conclusion and have multiple issues, which are outlined below.

Previous studies used genomics and high-resolution transcription assays to account for the link between histone acetylation and transcription. In the current study, using primarily mass-spec data the authors conclude that transcription doesn't affect transcription. It is important to note

that, at any given moment, only small subsets of genes are actively transcribing, and the impact of the inhibitors employed in this study is likely concentrated on these actively transcribing genes within mESCs. Therefore, the changes in acetylation in nucleosomes could be masked by the basal acetylation of other transcriptionally inactive genes or genomic regions that wouldn't be affected by inhibition. Furthermore, it is unclear what fraction of non-chromatin-bound histones were enriched in the analyses presented, considering that acetylated histones/nucleosomes might be evicted during transcription initiation or elongation. As such acetylation of these histones might not be impacted by transcription inhibition. Unfortunately, the authors have not furnished data on the effects of the inhibitors on transcription. This absence of comprehensive genome-wide data creates challenges in establishing a definitive connection between acetylation levels and transcriptional activity. Despite the lack of genome-wide data, they conclude "These results show that the global acetylation landscapes remain unchanged after acute transcription inhibition". The authors do not show the acetylation landscape, rather they only show global acetylation. As stated above, it is not even clear what proportion of material (histones and non-histone proteins) used for analysis are chromatin-bound. One would expect the changes in acetylation upon transcription inhibition to be on chromatin-associated histones and non-histone proteins. Furthermore, since authors used transcription inhibitors that affect different stages of transcription, therefore, the changes are expected to be localized to those locations: promoters vs gene bodies for transcription initiation vs elongation defects.

The authors must present the following high-resolution data sets in order to substantiate their findings:

1. Comprehensive changes in acetylation patterns before and after transcription inhibition across the entire genome.
2. Alterations in transcription levels and chromatin structure both pre- and post-inhibition. The levels of histones must be measured at each location to account for transcription-dependent changes in nucleosome occupancies.

Reviewer #4:

Remarks to the Author:

The paper by Liebner et al takes advantage of the well-established MS-proteomics platform setup in the Choudhary's lab for the global analysis of the K-acetyl-proteome to try address the important basic research question whether protein/histone acetylation is cause or consequence of transcription.

The experimental design proposed to address this question is relatively linear, combining:

- The use of SILAC labelling of cultures cell lines;
- The use of small molecule inhibitors to block transcription through different mechanism of action;
- The use of a potent inhibitor of the K-acetyl-transferase CBP/p300 as a positive control of a perturbation capable to affect the acetyl-proteome both on histones and non-histone proteins;
- The acetyl-proteomics platform already successfully applied to other studies by the same group.

The experiment performed are clear, linearly and logically presented; the manuscript is very well written and clear to follow.

Nevertheless, there are some major points that this reviewer needs to highlight, as they raise concerns about the suitability of this manuscript for publication in Nature Communications, as "Article".

The major criticism regards the analysis of histone acetylation in dependence of transcription, which is -as the authors clearly and correctly state in the Abstract- the most important open question to be addressed, but that is however addressed less robustly/convincingly in this study. The biochemical/analytical strategy employed here (based on tryptic digestion of whole cell extracts followed by affinity enrichment of acetyl-K-peptides then analyzed by LC-MS/MS) is in fact very effective for the analysis of global, non-histonic K-methylation, which however is not the real focus of the question that the authors want to address.

The question debated in the epigenetics/gene expression field concerns, instead, the

causal/consequence relationship between transcription and acetylation of HISTONES. However, It is well known, in the field of histone MS- proteomics, that Trypsin digestion is not ideal to analyze comprehensively and robustly histone modifications by MS, and that alternative approaches of sample preparation, histone digestion and analysis are required to both capture combinatorial PTMs and to be able to carry out reliable quantification of unmodified/modified sites.

Specifically, due to the high number of lysines and arginines present in histone sequences, trypsin digestion is not usually ideal for histone PTM analysis. Trypsin digestion, in fact, generates peptides too short for detection and that contain miscleavages due to the presence of modifications.

How did the authors deal with these issues, particularly for quantification?

The authors do not provide sufficient information on how the histone acetylation analysis has been carried out, explaining how they addressed with these issues, particularly for quantification.

Also, the results on histones displayed in Figure 3, consider all K-acetylation individually, suggesting that the analysis of multiply-acetylated peptides was not carried out. This is a strong limitation, given the, especially for specific core histone regions, the concurrent acetylation of multiple sites has historically and experimentally been associated mechanistically to transcriptional states and transitions.

This is, for instance, the case of:

- histone H4 N-terminal tail, with the concurrent tri- or tetra acetylation of K16/K12/K8 and K5;
- histone H3 with concurrent hyperacetylation of K9 and K14 and/or K18 and K23.

Hence, the workflow used in this study, while working well for global acetylation analysis, is not ideal for a thorough inspection and profiling of combinatorial acetylations on histones, which instead would be essential to address the question outlined.

The authors should not only provide more details on how the acetylation analysis on histones given the experimental data, but they should include the relative quantitation of the multiply acetylated peptides in response to the different treatments.

More information should be provided also regarding various aspect of the data analysis, for instance:

1) how were the ratios originating from the different biological replicates used to produce the final results?

2) how was the cut-off (=2) for defining differential regulation chosen? Why no statistics was applied?

3) were different histone isoforms quantified separately or were they collapsed? This is another important point, in light of the fact that various histone isoforms (e.g. H3.1 vs H3.3 have been linked to chromatin in distinct transcriptional states and should be analyzed, possibly, individually)

One last consideration is that -of the two alternative models of acetylation as cause of consequence of transcription, only the latter model has been (partially) addressed by the authors while no attempt to address the former option through a similar experimental design is present.

Hence, with the current set of experiments/analyses (even upon the improvement of analysis of histone acetylation aforementioned), the manuscript seems to fit more to a "letter" format rather than to a "full article" one, as its information content and message are not comparable to those presented in the "articles" usually published by the journal. Since, apparently, the "Letter" is not an optional format for Nat Communications, this reviewer has major concern about the overall appropriateness of this submission.

Point-by-point response to reviewers' comments

We thank the reviewer for evaluating the manuscript and for providing constructive feedback. We have performed several additional experiments and analyses, and the new results validates and further support the original conclusions. Below, we first respond to general comments, and then provide a point-by-point response to specific comments.

Response regarding bulk versus restricted changes in histone acetylation

Reviewers #1 and 3 commented that transcription inhibition may affect histone acetylation locus-specifically, without altering global levels of acetylation. Therefore, bulk level analysis of histones is not ideal for analyzing the impact of transcription inhibitors on acetylation. This appears a misunderstanding.

1. Contrary to the reviewer's assumption, two studies analyzed impact of transcription inhibition on histone acetylation, and both showed that transcription inhibition reduced **bulk** histone acetylation. Indeed, acetylation is reduced at virtually all sites analyzed. Thus, the idea is that transcription inhibition reduces acetylation of **all** histone sites, and **in bulk**. This is explicitly stated by Martin et al. "*Indeed, we found that inhibition of transcription by actinomycin D, which inhibits transcription initiation, as indicated by loss of Ser5p (Fig. 1d), and transcription elongation in mESCs resulted in the loss of H3K9ac and H3K27ac in bulk histones (Fig. 1d, e).*" As shown in the figure below, transcription inhibition reduced acetylation of all acetylation sites analyzed (Reviewer Figure 1, copied from PMID: 33431884).

Reviewer Figure 1: Shown is the relative decrease in the indicated chromatin marks in *S. cerevisiae* (left panel) cells treated with solvent control (EtOH) or transcription inhibitor (1,10-pt). Please note that acetylation of all analyzed histone sites is globally decreased following transcription inhibition. Of note, the decrease in H3K27ac is modest, likely because yeast lack an ortholog of CBP/p300, which is the primary acetyltransferase for this site in mammalian cells. In mouse ESC (right panel), H3K27ac and H3K9ac is strongly decreased after treatment with ActD. The Figure is directly copied from Martin et al. (PMID: 33431884).

2. Wang et al. analyzed H3K27ac by ChIP-seq, and showed that acetylation is decreased globally, both in promoters and enhancers (PMID: 35273399). Rather than revealing locus-specific regulation of H3K27ac, the H3K27ac ChIP-seq analysis merely confirmed bulk decrease in acetylation, as observed in immunoblotting.

To directly address reviewer's point, we analyzed change in H3K27ac (the same mark that was analyzed by Wang et al.) by ChIP-seq, after inhibiting transcription by ActD or NVP-2. Consistent with our mass spectrometry data, we found no appreciable decrease in H3K27ac in promoters or distal regions (new Fig. 5, see below).

Of note, we have used the same technique, H3K27ac ChIP-seq, and found that CBP/p300 inhibition rapidly (within 15 min) reduce genome-wide H3K27ac, and multiple sites in histone H2B (PMID: 37024579), demonstrating that the lack of H3K27ac decrease by transcription inhibition in our ChIP-seq data is not due to the inability of our experimental approach in detecting H3K27ac changes.

Fig. 5

Supporting our conclusions, and contradicting data from Martin et al. and Wang et al., two other publications, from independent groups, have shown that H3K27ac is not reduced after transcription inhibition. One study showed that acute depletion of MED14 globally reduce transcription, but without a notable decrease in H3K27ac (PMID: 32483291). Another study analyzed H3K27ac by IF and ChIP-seq and found no decrease in H3K27ac by triptolide (PMID: 32243828).

The reviewers appear to imply that transcription inhibition may regulate histone acetylation in some regions but not in others. We found no report showing that global transcription inhibition by any of the used inhibitors reduce acetylation of histones in a locus-specific manner, without altering bulk acetylation. We may have missed to find relevant literature. If the reviewer's assertion is based on concrete data in the literature, we kindly request them to share the relevant references with us so that we can look into them and respond accordingly.

Although we did not find any evidence to supporting the reviewer's assertion, we acknowledge this point in the revised manuscript and included the following text. *"In our analyses, acetylation of proteins, including histones, is analyzed in bulk. Hypothetically, it is possible that transcription inhibition decreases acetylation in some genomic regions, but this loss is compensated by an increase in acetylation in other regions. However, this rationale fails to explain the discrepancy with prior studies, which showed that acetylation of all analyzed histone sites was decreased in transcription inhibited cells, with little evidence for locus specific regulation in ChIP-seq analyses. It should be noted that different histone sites are acetylated by different acetyltransferases. It is difficult to conceive that transcription inhibition causes re-localization of all acetyltransferases to different chromatin regions such that acetylation is reduced in some regions and almost perfectly balanced by increased acetylation in other regions. Also, in our H3K27ac ChIP-seq data, we found no evidence of this. Nonetheless, we could not rule this possibility out entirely."*

Reviewer #1

In this manuscript Liebner et al aimed address the effect of transcriptional inhibition on histone acetylation levels by performing SILAC MS upon acute transcriptional inhibition. Recent evidences show that inhibition of transcription leads to reduced level of histone acetylations at transcription sites and transcription shapes histone acetylation pattern at expressed genes. Authors conclude that Acetylation on histones are not mere consequence of transcription and acetylation independent of transcription. Although authors findings are interesting and useful contribution to the field. However, measuring bulk acetylations using mass spec is not sufficient to conclude that acetylation is not mere consequence of transcription. I would encourage authors to perform additional experiments suggested below which will further strengthen their conclusions.

We thank the reviewer for evaluating the manuscript and for providing constructive feedback. We are encouraged that the reviewer found our work “interesting and useful contribution to the field.” We sincerely appreciate reviewer’s constructive feedback. Following reviewer’s suggestions, we have performed additional experiments and revised the text to clarify the points raised.

Major comments:

1. Assaying the bulk level of histone acetylations is not ideal to investigate the effect of transcription on histone acetylation at the transcription sites. What about the acetylation level at sites outside the transcribing genes, at regulatory elements. Can reduced histone acetylation at genes compensated by increased levels outside the genes? Chromatin and histone acetylation level at regulatory elements and transcription start sites is highly dynamic, it is important to assay altered level of histone acetylations at different genomic features.

In response to the reviewer’s comment, we analyzed changes in H3K27ac after transcription inhibition and found no decrease in acetylation. Our results are in agreement with ChIP-seq data of two other studies showing H3K27ac is not reduced by global transcription inhibition by triptolide treatment (PMID: 32243828), or acute depletion of MED14 (PMID: 32483291).

For a detailed response to this comment, we kindly request the reviewer to the introductory part of this rebuttal (see pages 1-2). In brief, prior claims that transcription is required for histone acetylation is based on the finding that transcription reduces histone acetylation in bulk, rather than affecting acetylation in a locus-specific manner.

2. Controls missing to demonstrate that triptolide treatment is indeed inhibiting transcription within 2 hours and at the used concentration.

1. To address this point, we analyzed changes in nascent transcription after triptolide treatment (2h). These analyses confirm that transcription is robustly inhibited (Extended Data Fig. 1).
2. Please note that all transcription inhibitors used here have been shown, but numerous studies, to globally reduce transcription in all mammalian cell lines tested. It may have escaped reviewer’s attention, but we showed that all three of the inhibitors virtually abrogated H2BK120Ub in mESC (Figure 4b). Ubiquitylation of histone H2BK120 (H2BK120Ub) is a well-established surrogate of ongoing transcription. Of note, unlike transcription inhibitors, H2BK120ub is not appreciably decreased after CBP/p300 inhibitor A-485, providing a useful control. In the same assay, H3K27ac was reduced by A-485, but not by transcription inhibitors, showing that A-485 and transcription inhibitors distinctly reduce H3K27ac and H2BK120Ub.

3. Can authors perform time course experiment by inhibiting transcription at different time point and assay the level of histone acetylations and association with transcription?

A time-course analysis is useful in two circumstances: (1) to identify change in modification that occur transiently and restored later, and (2) to quantify the kinetics of modifications that do show a change. In the context of transcription-induced change in histone acetylation, we do not have evidence for either of them.

Prior work showed that acetylation of H3K27ac is completely decreased within 1 hour and remained inhibited until 4 hours, showing that transcription inhibition causes rapid and sustained decrease in histone acetylation. Based on these reports, we chose the 2h time point for our analyses. It is not conceivable to us that acetylation is decreased at 1h and 4h time points, but not at 2 hours. Contrary to prior reports, we do not see a rapid decrease in acetylation after transcription inhibition. Because acetylation sites show no change in this timeframe (2h). We do not claim that transcription has not impact on acetylation after long-term treatment. In long-term treatments, it is complicated to disentangle direct and indirect effects of transcription inhibition on acetylation. If transcription is inhibited for too long, it will inevitably affect expression of genes and proteins, and make it impossible to determine whether change in acetylation is directly regulated by transcription, or indirectly regulated by change in mRNA/protein expression. We clearly discuss in the manuscript that our conclusion is that acetylation is not measurably reduced within 2h, and we do not suggest that acetylation will remain unchanged after long-term transcription inhibition.

In the conditions where acetylation is dynamically regulated, we agree that time-course is useful to understand the kinetics. It is for this reason, we performed time-course experiments after the inhibition of CBP/p300 activity (PMID: 29804834). However, because we do not observe a measurable, rapid change in histone acetylation after 2h of transcription inhibition, we do not see a good reason to analyze transcription at earlier time points. If the reviewer can elaborate how time-course experiments will be helpful under these circumstances, we will be happy to consider his/her suggestions.

4. Authors lab has recently published their work on histone acetylations by performing ChIP-seq for histone acetylation, since the lab has expertise and I expect authors to perform quantitative ChIPseq for H3K27ac and analyse the level at genes and intergenic sites.

Our group have used both mass spectrometry and ChIP-seq to quantify changes in histone acetylation. The choice of the method is based on the question being addressed- for studies where we need to accurately quantify bulk changes in acetylation, mass spectrometry is clearly the method of choice. We genuinely believe that, for bulk quantification of PTMs, ChIP-seq is not as accurate as mass spectrometry. On the other hand, if the goal is to study locus-specific changes in specific acetylation mark, we recognize that ChIP-seq is absolutely necessary.

As pointed out above, transcription inhibitors were reported to reduce histone acetylation in bulk, and across the genome, not locus-specifically (PMID: 33431884, PMID: 35273399). Therefore, in the original submission, we felt that performing ChIP-seq is uninformative as it provides no additional information that is not provided by bulk analysis of histones.

Regardless, following reviewer's comment, we have performed ChIP-seq for H3K27ac (PMID: 33431884, PMID: 35273399). Consistent with our mass spectrometry data, we found no measurable decrease in H3K27ac in these data.

5. Manuscript focus on H3K27ac, there are many acetylations on histone that are possibly altered upon perturbing transcription. For e.g. Like H3K36me3, H4K16ac is enriched on the gene bodies in the transcribed genes. It would be good to check if this marks is altered when transcription is inhibited.

Firstly, we would like to emphasize that our analyses are not limited to histones, or H3K27ac. Instead, we performed proteome-scale analyses and quantified >20,000 acetylation sites, including dozens of sites in histones. In these unbiased analyses, we found no indication of site-specific regulation of histones, and all sites, including H4K16ac, remain virtually unchanged.

Secondly, we would like to mention that, out of 4 main figures, only one is related to H3K27ac. The reason why we examined H3K27ac is that is the only mark that was analyzed in both the prior studies, which concluded that ongoing transcription was required for histone acetylation.

Thirdly, we wish to clarify that focus of our work is on acetylation. We make no claims about the effect of transcription inhibitors on other chromatin modifications. Nonetheless, for reviewer's information, Wang et al. reported that H3K36me3 is not reduced by transcription inhibition, even though this mark is known to be deposited co-transcriptionally (PMID: 35273399). They rationalized this by the fact that H3K36me3 has long half-life, and therefore, its levels do not decrease (within 4h) after transcription inhibition.

Minor comments: Abstract is too brief, please include organism and methodology used.

We would have certainly considered including these details in the abstract. However, the journal limits abstract to maximum 150 words, and the current abstract is already 150 words long, making it challenging to include details on methodology and organism in the abstract.

Reviewer #2

Many thanks for the opportunity to review your manuscript. I enjoyed being able to take some time to read it and make myself at least superficially familiar with what seems to be an exciting and vibrant field. The introduction is very well presented and presents the apparently conflicting evidence in the sequentially of histone acetylation and transcription clearly and comprehensively. More generally though, while on its own it is of interest to study the relationship between global protein acetylation and levels of transcriptional activity, it is not clear to me what the relevance is of acetylation of non-histone proteins to a discussion on causality of histone acetylation and transcription. I wonder if the penultimate paragraph of the introduction would benefit from highlighting the possibility of recruitment of HATs to transcriptionally active loci, as mentioned earlier in the introduction, as a mechanism of histone acetylation as specific from global acetylation.

We thank the reviewer for evaluating the work and providing helpful feedback. We sincerely appreciate their expert feedback.

"...it is not clear to me what the relevance is of acetylation of non-histone proteins to a discussion on causality of histone acetylation and transcription."

We analyze and discuss the acetylation of both histone and non-histone proteins for the following reasons:

1. It is suggested that ongoing transcription is required for the activation of acetyltransferases or their recruitment to the chromatin (PMID: 33431884, 35273399).

2. The same acetyltransferases that acetylate histones also acetylate most non-histone proteins in the cell (PMID: 35273399).
3. In addition to histones, acetylation of non-histone proteins is implicated in the regulation of gene transcription (PMID: 9288740, 23431171, 11509556, 11250901, 27613418).

Given that acetylation of both histone and non-histone proteins is implicated in transcription regulation, we believe it is important to investigate and discuss the role of ongoing transcription in the acetylation of both histone and non-histone proteins.

“I wonder if the penultimate paragraph of the introduction would benefit from highlighting the possibility of recruitment of HATs to transcriptionally active loci...”

It is well-established that all canonical acetyltransferases are recruited to transcriptionally active regions. Since this is a well-recognized fact, we believe that repeating it in the discussion may not be necessary. However, if the reviewer finds it helpful, we will happily comply and revise the text accordingly.

Methods & Results

I was specifically asked to comment on the imaging cytometry, and I will mainly restrict my specific comments to that part of the data as the SILAC mas spec is not an expertise of mine. I appreciate though from a brief look at the data that minimal, if any, changes were observed in global protein acetylation following inhibition of transcription, in contrast to convincing and time dependent changes following acetyltransferase inhibition. Figure 3 appears to present strong direct evidence that inhibition of transcription does not alter histone acetylation. Apologies if I missed this but why was the transcription inhibition done at a single time point of two hours but the CBP/p300 inhibition at multiple timepoints up to 1 hour?

We want to clarify that the quantification of CBP/p300 inhibitor-induced changes in acetylation was conducted in a separate study (PMID: 29804834), which was completed prior to the start of this project. In this work, the rationale for analyzing acetylation at multiple time points was to quantify the in vivo turnover rate (half-life) of CBP/p300-regulated sites.

The choice of the transcription inhibition time point is based on two recent publications demonstrating a significant reduction in histone acetylation (including H3K27ac) within 1 to 4 hours (PMID: 33431884, PMID: 35273399). Therefore, for consistency, acetylation changes were quantified after 2 hours of transcription inhibitor treatment.

Data from CBP/p300 inhibitor experiments are solely used here to demonstrate the suitability of our approach in quantifying acetylation site changes and to illustrate the dynamic regulation of acetylation. We acknowledge that it is unnecessary to present data from all time points. To avoid confusion, we have removed the data from multiple time points and included data from the 1-hour treatment.

The image acquisition and analysis are suited for the intended quantitative analysis, with specific reference made in the methods to the set image analysis parameters fixed for all images. The methods would also benefit from confirmation that all parameters set also fixed for image acquisition. Additionally, it would be useful to see the associated DAPI images in parallel in figure 4 (as confirmation of the presence of cells) and an illustration of the masking and quantification strategy in supplementary data.

We thank the reviewer for this suggestion. We have revised the text to include the suggested details and included DAPI images in Figure 4. We confirm that same fixed parameters were used for image acquisition and processing. Illustration of the masking and quantification strategy is now provided in Extended Data Fig. 3.

The presented data (Figure 4) shows a clear retention of signal detected with the H3K27ac antibody following exposure to transcriptional inhibitors.

We fully agree with the reviewer's assessment.

Lines 291 and 292 refer to a supplementary table with details of the primary antibodies used. Apologies if I'm missing it but I can't find it. This is critical to assessing the quality of the IF analysis. I would also like to see experimental assessment of the specificity of the antibodies used for IF. Specificity of the H3K27ac antibody for H3K27ac against other acetylated proteins is essential to support the claim that CBP/p300 catalysed H3K27 acetylation is independent of transcription, to demonstrate the retained staining isn't due to non-specific detection of other lysine acetylation. I think the robustness of the conclusions from the IF needs to be toned down, especially in the context of the absence of data demonstrating the specificity of the H3K27ac antibody. It cannot be assumed based on the data currently presented that the Ab is detecting H3K27ac, or detecting H3K27ac alone. Line 143 to 156 I feel is not justified. While the IF results demonstrate convincingly that acetylated proteins are present in the presence or absence of transcription, they do not demonstrate that acetyltransferases continue to activate histones in the absence of ongoing transcription.

Firstly, we sincerely apologize that the information on antibodies was not readily accessible to the reviewer. This information was provided in the indicated table, but under the tab "QIBC." For convenience to readers, in the revised manuscript, we have directly stated this information within the Methods section, describing IF analyses.

Secondly, we would like to clarify that the H3K27ac antibody used here is not a new antibody. The monoclonal antibody used in our analyses is extensively characterized and has been used in dozens of high-ranked publications, including for immunofluorescence and immunohistochemistry (PMID: 34569155, PMID: 33852836, PMID: 36669472). Also, please note that H3K27ac signal is decreased (both in IF and mass spectrometry) after the inhibition of CBP/p300, which is the known acetyltransferase for this site. If the reviewer feels that it is necessary to re-confirm the specificity, we will perform the required controls.

Thirdly, in response to reviewer's comment, we have used another H3K27ac antibody (Rabbit monoclonal, clone #D5E4), Cat #8173, Cell Signaling Technology) to independently confirm the original results (Extended Data Fig. 2). This antibody has also been used in numerous prior ChIP-seq studies, as well as for immunofluorescence-based analyses (PMID: 34370796; PMID: 25409661; <https://www.biorxiv.org/content/10.1101/2023.08.15.553204v1.full.pdf>).

Extended data Fig. 2

a

b

We would like to mention that, consistent with our data, another study analyzed H3K27ac by IF and ChIP-seq and found no decrease in H3K27ac by triptolide (PMID: 32243828). Another study shows that pre-treating cells with DRB (5,6-dichloro-1-β-D-ribofuranosylbenzimidazole), a transcription elongation inhibitor, abrogated ligand-induced activation of a gene, but it did not abrogate ligand-induced acquisition of H3K18ac, H3K27ac, and histone H4 (PMID: 21131905).

Finally, we would like to clarify that our conclusion that “acetyltransferases continue to activate histones in the absence of ongoing transcription” is based on the combined evidence from both IF and mass spectrometry analyses. Our mass spectrometry data show that acetylation of hundreds of histone and non-histone sites is reduced rapidly after CBP/p300 inhibition, but not after transcription inhibition. In IF analyses, acetylation of H3K27ac is not reduced in cells treated with transcription inhibitors alone, but if they are co-treated with the CBP/p300 inhibitor, H3K27ac is reduced. This shows that CBP/p300 remains active in cells treated with transcription inhibitors. In response to reviewer’s comment, we have revised the text. Instead of generalizing this result to all acetyltransferases, we specifically mention that CBP/p300 remains active in transcription inhibited cells, and this result is adequately supported by the provided data. *“We further show that CBP/p300 continues to acetylate H3K27ac in transcription inhibited cells, indicating that the recruitment and activation of these acetyltransferases is uncoupled from the act of transcription. Since most acetylation appears to occur independently of ongoing transcription. Therefore, histone acetylation should not merely be considered a consequence of ongoing transcription.”*

We would like to mention that activity of CBP/p300 in transcription inhibited cells is also evident in our recent work (PMID: 37024579). As it can be seen in the below genome-browser tracks, transcription inhibition (NVP-2, 2h) results in an increase in histone H2BK20ac, which is catalyzed by the same enzyme that catalyze H3K27ac. Remarkably, H2BK20ac increased after transcription inhibition, and only in actively transcribed gene body region, not in the promoter upstream region. The specific increase in actively transcribed regions, after transcription inhibition, further suggests that CBP/p300 must remain active after CBP/p300 inhibition. As a note, within this same work, we demonstrated that H2BK20ac is decreased within 5-15 minutes of CBP/p300 inhibition.

Reviewer #3

This study by Choudhary and colleagues examined changes in histone acetylation levels after acute inhibition of transcription in mouse embryonic stem cells by mass-spec analysis. They used three different transcription inhibitors, which affected initiation, elongation, or RNA Pol II release. In contrast to previously published studies, some of which are referenced in the current studies, they found that transcription inhibition did not significantly reduce acetylation. As expected, inhibiting CBP/P300 activity resulted in significant reductions in acetylation at various sites, including the core histones. Based on these studies they conclude histone acetylation is uncoupled from the transcription. The authors have tried to address an important question—the link between histone acetylation and transcription. However, their approach and results are inadequate in supporting their conclusion and have multiple issues, which are outlined below.

We thank the reviewer for assessing our work and providing critical feedback. We are happy to note that the reviewer recognizes the importance of the question addressed in our work. Reviewer also raise several points, which we clarify below.

Previous studies used genomics and high-resolution transcription assays to account for the link between histone acetylation and transcription. In the current study, using primarily mass-spec data the authors conclude that transcription doesn't affect transcription. It is important to note that, at any given moment, only small subsets of genes are actively transcribing, and the impact of the inhibitors employed in this study is likely concentrated on these actively transcribing genes within mESCs. Therefore, the changes in acetylation in nucleosomes could be masked by the basal acetylation of other transcriptionally inactive genes or genomic regions that wouldn't be affected by inhibition. Furthermore, it is unclear what fraction of non-chromatin-bound histones were enriched in the analyses presented, considering that acetylated histones/nucleosomes might be evicted during transcription initiation or elongation. As such acetylation of these histones might not be impacted by transcription inhibition. Unfortunately, the authors have not furnished data on the effects of the inhibitors on transcription. This absence of comprehensive genome-wide data creates challenges in establishing a definitive connection between acetylation levels and transcriptional activity. Despite the lack of genome-wide data, they conclude "These results show that the global acetylation landscapes remain unchanged after acute transcription inhibition". The authors do not show the acetylation landscape, rather they only show global acetylation. As stated above, it is not even clear what proportion of material (histones and non-histone proteins) used for analysis are

chromatin-bound. One would expect the changes in acetylation upon transcription inhibition to be on chromatin-associated histones and non-histone proteins. Furthermore, since authors used transcription inhibitors that affect different stages of transcription, therefore, the changes are expected to be localized to those locations: promoters vs gene bodies for transcription initiation vs elongation defects.

The reviewer's main concern is that transcription inhibition may impact acetylation in restricted genome regions, and these changes will not be observed in bulk analysis of histones. This appears a misunderstanding; we are not aware of any genome-scale analyses showing that transcription inhibition by Actinomycin D, NVP-2, triptolide, or any other global transcription inhibitor results in locus-specific changes in acetylation. If the reviewer is aware of any genome-scale analyses showing that different transcription inhibitors distinctly affect acetylation, or that they affect acetylation in specific genomic regions, we kindly request the reviewer to share the relevant references with us so that we can more specifically respond to them.

Because of the compound nature of the above comment, we address it in parts.

1. “Previous studies used genomics and high-resolution transcription assays to account for the link between histone acetylation and transcription. In the current study, using primarily mass-spec data the authors conclude that transcription doesn't affect transcription.”

Following reviewer's feedback, we have performed ChIP-seq for H3K27ac after Actinomycin D and NVP-2 treatment. Consistent with our mass spectrometry data, we found no measurable decrease in H3K27ac after transcription inhibition, showing that lack of acetylation decrease is not due to inability of mass spectrometry to detect bulk decrease in histone acetylation. Our findings are inconsistent with those published by Martin et al. and Wang et al. (PMID: 35273399), but they concur with the data published by Rosencrance et al. (PMID: 32243828) who wrote that “... *we asked whether we could decouple gene expression from chromatin composition by inhibiting RNA polymerase II initiation with triptolide (Jonkers et al., 2014, Titov et al., 2011, Wang et al., 2011). Treating TC-797 cells for 4 h with 500 nM of triptolide resulted in clearance of RNA polymerase II from promoters, gene bodies (Figure 7A), and megadomains (Figure S7A) as well as genome-wide loss of transcription (Figure 7B). However, BRD4-NUT or H3K27ac nuclear foci persisted in the absence of transcription (Figure 7C). H3K27ac increased genome-wide, including at megadomains (Figure 7D), indicating that chromatin hyperacetylation did not depend on RNA polymerase II initiation or transcription.*”

Further supporting our data, acute depletion of MED14 globally reduce transcription, but H3K27ac remain unchanged (PMID: 32483291).

Together, our new ChIP-seq data, results from prior studies, support the conclusion that H3K27ac is not changed by transcription inhibition. Therefore, the failure to reproduce the findings from Martin et al. and Wang et al. is not due to the use of different technologies (ChIP-seq versus mass spectrometry).

Of note, Wang et al. (PMID: 35273399) reported that triptolide treatment globally reduce H3K27ac, both in promoters and in enhancers, where H3K27ac is prominently enriched. Thus, there is also no evidence in their data that transcription affect histone acetylation, without reducing global decrease, as the reviewer appears to imply.

2. “It is important to note that, at any given moment, only small subsets of genes are actively transcribing, and the impact of the inhibitors employed in this study is likely concentrated on these actively transcribing genes within mESCs. Therefore, the changes in acetylation in

nucleosomes could be masked by the basal acetylation of other transcriptionally inactive genes or genomic regions that wouldn't be affected by inhibition.”

The reviewer suggests that: (1) only a small fraction of genes is actively transcribing, (2) transcription inhibition would only affect acetylation in actively transcribed regions, and these changes in acetylation could be masked by acetylation in other transcriptionally inactive genes or genomic regions that would be not impacted by transcription inhibitors.

Following reviewer's comment, we also checked our data for the presence of acetylation in actively transcribed and low/no transcribed genes and change in acetylation after transcription inhibition in different genomic regions. We find that mESCs expresses over half of the known genes in the mouse genome. H3K27ac marks most of actively transcribed genes, while most low or not transcribed genes lack it, and those that are marked, acetylation is very low (Fig. 5b, see page 2). After transcription inhibition H3K27ac is not increased in low or no expressed genes (Fig. 5a,c, see page 2). This shows that the lack of global H3K27ac in our mass spectrometry analyses is unlikely due to genomic re-distribution of H3K27ac.

Of note, if histones were already acetylated in transcriptionally inactive genes, that would already mean that ongoing transcription is not required for acetylation of histones, which will already nullify the current models that transcription is required for histone acetylation.

We would like to further mention that H3K27ac ChIP-seq data by Wang et al. also did not find any evidence of genomic re-distribution of H3K27ac; instead, they found that H3K27ac is reduced in bulk (PMID: 35273399).

3. “...authors used transcription inhibitors that affect different stages of transcription, therefore, the changes are expected to be localized to those locations...”.

We are uncertain about the basis of this idea. As shown in the above figure (see page 2), we did not find any difference in H3K27ac regulation by Actinomycin D or NVP-2, and prior studies (PMID: 35273399; PMID: 32243828) presented conflicting effect of transcription inhibition on H3K27ac. However, none of the studies found any evidence of locus-specific regulation of histone acetylation. To the best of our knowledge, no study has shown that transcription inhibition by Actinomycin D, NVP-2, or triptolide impact histone acetylation in a locus or region-specific manner. Contrary to this idea, prior research has reported that acetylation affected globally by actinomycin D and triptolide. This was the original title of the work by Martin et al. was “The majority of histone acetylation is a consequence of transcription” “<https://www.biorxiv.org/content/10.1101/785998v1.abstract>”

4. “Furthermore, it is unclear what fraction of non-chromatin-bound histones were enriched in the analyses presented, considering that acetylated histones/nucleosomes might be evicted during transcription initiation or elongation. As such acetylation of these histones might not be impacted by transcription inhibition.” “One would expect the changes in acetylation upon transcription inhibition to be on chromatin-associated histones and non-histone proteins.”

Again, this notion appears inconsistent with the published data, which showed that transcription inhibition reduces acetylation globally, within promoters and enhancers, and all acetylation sites analyzed. Using whole cell extract, prior work showed that acetylation is reduced in bulk histones. Therefore, we used whole cell extracts and quantified acetylation

bulk proteins. For detailed response on this point, please see the introductory part of this rebuttal (pages 1-2), and the figure shown there.

In our study, proteins were extracted by directly lysing the cells on plate in the buffer containing 2% sodium deoxycholate and boiling the lysate at the lysates at 95°C for 10 minutes. This approach robustly extracts all chromatin and non-chromatin bound proteins. Therefore, our approach can quantify acetylation changes in bulk proteins, regardless of whether the proteins were bound to chromatin or not.

However, using the same inhibitors (Actinomycin D and triptolide), and same cell line, we do not observe a decrease in acetylation, although we do observe strong decrease histone ubiquitylation of histone H2B. Also, we observe strong decrease in H3K27ac by CBP/p300 inhibition, showing that our approach is capable of quantifying changes in acetylation.

The reviewer suggests that acetylation histones may be evicted during transcription initiation or elongation. If this was the case, then inhibition of transcription should prevent eviction of acetylated histones, and thus, cause an increase in acetylation, not a decrease in acetylation.

This was directly demonstrated in our recent work (PMID: 37024579). During elongation, a nucleosome is split, and one dimer of H2A/H2B is likely exchanged, while the hexamer containing second dimer of H2A/H2B, and both the dimers of H3/H4, is redeposited (PMID: 25650798). Because of this, transcription inhibition by NVP-2 selectively increase (not decrease) acetylation of H2B in actively transcribed genes. The only possible reason why we may not see a decrease in acetylation is that chromatin-associated proteins are also acetylated on the same residues in their non-chromatin bound state, in the absence of transcription inhibition. If so, the amount of non-chromatin-bound acetylated protein pool must be so large that any decrease in the acetylation of chromatin-bound protein pool is masked by acetylation of non-chromatin-bound pool. We do not believe that this is likely, but we could not rule this possibly out. We revised the manuscript to acknowledge this possibility.

5. “The authors do not show the acetylation landscape, rather they only show global acetylation.”

We apologize for this unclarity. We would like to clarify that the word “landscape” is used for different meanings in different fields. In the field of proteomics, this word is frequently used for analyses that spans on the proteome-wide scale (i.e. not restricted to specific proteins). We used this word because our analyses quantified over 20,000 sites across the proteome.

We recognize that in the field of genomics/epigenetics, the word “landscape” is used to refer to analyses performed on the genome-wide scale, even if such analyses may involve analysis of a single acetylation sites. We have re-worded the text to better clarify the context in which this word is used in our work.

The authors must present the following high-resolution data sets in order to substantiate their findings:

1. Comprehensive changes in acetylation patterns before and after transcription inhibition across the entire genome.

Following reviewer's suggestion, we have analyzed genome-wide changes in H3K27ac, without inhibiting transcription, and after inhibiting transcription by two different inhibitors: Actinomycin D and NVP-2. Consistent with our mass spectrometry and immunofluorescence data, we find no measurable change in H3K27ac in ChIP-seq.

2. Alterations in transcription levels and chromatin structure both pre- and post-inhibition. The levels of histones must be measured at each location to account for transcription-dependent changes in nucleosome occupancies.

We believe analysis of nucleosome occupancy only becomes necessary if acetylation of histones is reduced after a treatment. However, this is not the case here. In that case useful to measure nucleosome occupancy to determine whether the decrease results from the removal of the modification only, or removal of the nucleosome. Because of H3K27ac is not reduced, we sincerely see not see a point in analyzing nucleosome occupancy.

Even the studies that have reported decrease in histone acetylation after transcription inhibition reported that nucleosome occupancy or position was not changed. For example, Martin et al. (PMID: 33431884) wrote that: *"While no major changes to nucleosome occupancy or position were observed following the short transcription inhibition performed here, large decreases in histone acetylation were observed (Fig. 1f)."*

In mammalian cells, most H3K27ac is found in enhancer regions (PMID: 21106759). Wang et al. showed that triptolide treatment strongly reduced H3K27ac in enhancers, without reducing the nucleosome occupancy (PMID: 35273399).

We wish to clarify that our work makes no claim on the role of transcription on chromatin structure. Therefore, studying the role of transcription in chromatin structure lies besides the scope of this work.

Reviewer #4

The paper by Liebner et al takes advantage of the well-established MS-proteomics platform setup in the Choudhary's lab for the global analysis of the K-acetyl-proteome to try address the important basic research question whether protein/histone acetylation is cause or consequence of transcription.

The experimental design proposed to address this question is relatively linear, combining:

- The use of SILAC labelling of cultures cell lines;
- The use of small molecule inhibitors to block transcription through different mechanism of action;
- The use of a potent inhibitor of the K-acetyl-transferase CBP/p300 as a positive control of a perturbation capable to affect the acetyl-proteome both on histones and non-histone proteins;
- The acetyl-proteomics platform already successfully applied to other studies by the same group.

The experiment performed are clear, linearly and logically presented; the manuscript is very well written and clear to follow. Nevertheless, there are some major points that this reviewer needs to highlight, as they raise concerns about the suitability of this manuscript for publication in Nature Communications, as "Article". The major criticism regards the analysis of histone acetylation in dependence of transcription, which is -as the authors clearly and correctly state in the Abstract- the most important open question to be addressed, but that is however addressed less robustly/convincingly in this study. The biochemical/analytical strategy employed here (based on tryptic digestion of whole cell extracts followed by affinity enrichment of acetyl-K-peptides then

analyzed by LC-MS/MS) is in fact very effective for the analysis of global, non-histonic K-methylation, which however is not the real focus of the question that the authors want to address. The question debated in the epigenetics/gene expression field concerns, instead, the causal/consequence relationship between transcription and acetylation of HISTONES. However, It is well known, in the field of histone MS- proteomics, that Trypsin digestion is not ideal to analyze comprehensively and robustly histone modifications by MS, and that alternative approaches of sample preparation, histone digestion and analysis are required to both capture combinatorial PTMs and to be able to carry out reliable quantification of unmodified/modified sites. Specifically, due to the high number of lysines and arginines present in histone sequences, trypsin digestion is not usually ideal for histone PTM analysis. Trypsin digestion, in fact, generates peptides too short for detection and that contain miscleavages due to the presence of modifications. How did the authors deal with these issues, particularly for quantification? The authors do not provide sufficient information on how the histone acetylation analysis has been carried out, explaining how they addressed with these issues, particularly for quantification.

We appreciate the reviewer's effort in reviewing the work, and for their feedback. We are pleased that the reviewer found that "The experiment performed are clear, linearly and logically presented; the manuscript is very well written and clear to follow." We acknowledge the concerns raised by the reviewer, which we believe may stem from misunderstandings due to unclear presentation or insufficient methodological details. In response to the reviewer's feedback, we have made efforts to enhance clarity by providing additional methodological details to better explain the steps involved in data processing and generating the final data tables. We apologize for any previous shortcomings in clarity and sincerely thank the reviewer for their valuable input.

Below, we clarify specific issues raised in the above assessment.

"The biochemical/analytical strategy employed here (based on tryptic digestion of whole cell extracts followed by affinity enrichment of acetyl-K-peptides then analyzed by LC-MS/MS) is in fact very effective for the analysis of global, non-histonic K-methylation, which however is not the real focus of the question that the authors want to address. The question debated in the epigenetics/gene expression field concerns, instead, the causal/consequence relationship between transcription and acetylation of HISTONES."

1. It is true that prior work has only analyzed the effect of transcription inhibitors on acetylation of sites in histones. In fact, the scope of prior work was limited to sites occurring in histones H3 and H4 only. We kindly request the reviewer to note that our work does not leave out histones. Rather, the work provides much deeper coverage of histone acetylation sites than any of the prior studies looking into transcription inhibition-induced changes in histone acetylation. In our analyses, we have quantified transcription inhibition-induced changes in acetylation of all four core histones, and our analyses include most of the sites that are studied in the epigenetics/chromatin field. Therefore, we kindly request the reviewer to recognize that by studying global acetylation, we are not leaving out the question that is interesting to colleagues in the epigenetics/chromatin/transcription field.
2. We also would like to stress that the role of acetylation in gene regulation is not restricted to histones. Acetylation of numerous non-histone proteins has been implicated in gene regulation. For example, it has been reported that acetylation regulates function of numerous transcription factors, including p53 (PMID: 9288740), HSF1 (PMID: 19229036), GATA1 (PMID: 9859997), FOXO1 (PMID: 16076959), and NF- κ B (PMID: 11533489). Acetylation of the transcription factors Ets1 (PMID: 28851877) and Twist (PMID: 24525235) is reported to promote the recruitment of transcription co-activator BRD4. Acetylation of RNA polymerase 2 CTD has been reported to regulate gene transcription (PMID: 24207025).

Given the extensive literature on the role of non-histone proteins we are unsure why it would not be interesting for the epigenetics/transcription community to know whether ongoing transcription is required for acetylation of non-histone transcription regulators.

We believe that it is important to clarify the role of ongoing transcription in both histone and non-histone proteins. We would emphasize that it is the same acetyltransferases that are shown to catalyze acetylation of histone and non-histone transcription regulators. There is no evidence/reason to believe that transcription specifically impact their activity towards histones. Therefore, it is important to reveal whether transcription selectively affect sites in histone or non-histone proteins. Extending the analyses beyond histones, expands the appeal of this work to readers beyond the transcription/epigenetic field. Just by analyzing histones, we could not establish whether transcription selectively impact acetyltransferase activity towards histones, or it also impact non-histone proteins. Also, acetyltransferases, such as CBP and p300, are strongly regulated by autoacetylation (PMID: 15004546), which is regulated highly dynamically (PMID: 17065153). These sites provide an indication about the activation state of these enzymes. By showing that autoacetylation sites in acetyltransferases remain unaffected by transcription inhibition implies that these enzymes likely remain active in transcription inhibited cells.

In brief, our work aims to address the broader question about the role of transcription in shaping acetylation site landscape in cells, but it does not exclude histones. Because of its more expansive scope, we believe that the work will be interesting to readers beyond those interested in acetylation of histones.

“However, it is well known, in the field of histone MS- proteomics, that Trypsin digestion is not ideal to analyze comprehensively and robustly histone modifications by MS, and that alternative approaches of sample preparation, histone digestion and analysis are required to both capture combinatorial PTMs and to be able to carry out reliable quantification of unmodified/modified sites. Specifically, due to the high number of lysines and arginines present in histone sequences, trypsin digestion is not usually ideal for histone PTM analysis. Trypsin digestion, in fact, generates peptides too short for detection and that contain miscleavages due to the presence of modifications. How did the authors deal with these issues, particularly for quantification? The authors do not provide sufficient information on how the histone acetylation analysis has been carried out, explaining how they addressed with these issues, particularly for quantification.”

Here, the reviewer raises two separate points: coverage of histone acetylation sites in trypsin-based MS proteomics, and the reliability of peptide quantification in our approach.

Regarding comprehensiveness of the coverage of acetylation sites, we recognize the limitation pointed out. Acknowledging this generic limitation, we would like to clarify that, to the best of our knowledge, this study presents one of the deepest analyses of acetylation changes in response to transcription inhibition, both for histone and non-histone sites. However, we do not claim that we have comprehensively quantified all sites.

In this respect, we kindly request the reviewer to note that the coverage of acetylation sites in prior studies is much smaller than our work. One study analyzed acetylation of a single site (H3K27ac), and other analyzed sites only in the N-termini of histone H3 and H4, and none of these studies analyzed acetylation sites in any combination. Furthermore, to the best of our knowledge, no study has shown that transcription inhibition affects acetylation of singly and multiply modified histone sites distinctly.

Regardless, we recognize reviewer's point and added the following text in the revised manuscript to explicitly acknowledge this limitation.

"We want to clarify that our approach involves the use of trypsin proteasome, and our chromatography set-up will likely not detect peptides that are either too short or too long (<5, or over ~30 amino acids). Therefore, our analyses are not comprehensive and there may be sites or specific combinations of sites in histones or non-histone proteins that are regulated by transcription inhibition but not quantified in our study. Notwithstanding this generic limitation, our analyses provide provides one of the deepest coverages of global acetylation and include a great majority of known sites in histones."

Regarding the reliability of quantification, with all due respect, we disagree with that the trypsin/SILAC-MS-based approach used here is not suitable for robust quantification of histone acetylation. The reviewer appears to conflate the issue of peptide identification with the issue of peptide quantification. In SILAC-based quantification, the quantified peptides are chemically identical, and singly and multiply modified peptides are quantified independently of each other. Therefore, whether a quantified peptide is singly or multiply modified has no bearing on the accuracy of quantification. The approach used here has been used by us and many other groups for quantifying lysine modification, including acetylation and ubiquitylation, in histones and non-histones (PMID: 17267393, PMID: 25751058, PMID: 34019788, PMID: 27913581). Indeed, the suitability of this approach is demonstrated in the data presented in this work. We show that acetylation is strongly and site-selectively reduced after the inhibition of CBP/p300, but not after the inhibition of transcription. Indeed, as it can be seen in the figure below, we quantified most of known acetylation sites in histones and acetylation of histone sites is decreased site-specifically, and in a time-dependent manner, after CBP/p300 inhibition. Because exact same approach is used for quantifying acetylation after these perturbations (CBP/p300 inhibition, transcription inhibition), lack of histone acetylation reduction in transcription inhibitor treated samples could not be attributed to problems in quantification. We are confident that our approach is suitable for quantifying acetylation in both histone and non-histone proteins.

Because the comment on peptide quantification is repeated below, we address this in the comment below.

Also, the results on histones displayed in Figure 3, consider all K-acetylation individually, suggesting that the analysis of multiply-acetylated peptides was not carried out. This is a strong limitation, given the, especially for specific core histone regions, the concurrent acetylation of multiple sites has historically and experimentally been associated mechanistically to transcriptional states and transitions. This is, for instance, the case of:

- histone H4 N-terminal tail, with the concurrent tri- or tetra acetylation of K16/K12/K8 and K5;
- histone H3 with concurrent hyperacetylation of K9 and K14 and/or K18 and K23.

Hence, the workflow used in this study, while working well for global acetylation analysis, is not ideal for a thorough inspection and profiling of combinatorial acetylations on histones, which instead would be essential to address the question outlined.

The reviewer correctly states that histone acetylation sites are quantified in peptides harboring one or more acetylation sites. Therefore, data should be made available for all forms of modified peptide quantified. We fully agree with this point. Following reviewer's suggestion, we have included a separate table listing all acetylated peptides quantified in histones.

In Figure 3, acetylation site ratios for individual sites were collapsed for the following reasons. (1) Histone acetylation is mostly studied using site-specific antibodies, which by design, recognize individual sites and do not inform on the combinatorial occurrence of other modifications. (2) We are not aware of any concrete evidence that transcription inhibition differently affects histone acetylation sites present in different combinations. (3) Presenting ratios at individually sites help reduce the data complexity such that changes in all histone acetylation sites can be shown in a single figure.

The authors should not only provide more details on how the acetylation analysis on histones given the experimental data, but they should include the relative quantitation of the multiply acetylated peptides in response to the different treatments. More information should be provided also regarding various aspect of the data analysis, for instance:

1) how were the ratios originating from the different biological replicates used to produce the final results?

To produce the final results, acetylation site ratios were normalized by the corresponding proteome measurement of each experiment, and any sites with a localization probability of < 0.9 were discarded. For site quantification, we required a minimum 3 ratios for actinomycinD and NVP2 experiments, a minimum of 2 ratios for triptolide. For the sites passing these criteria, median ratios were calculated and log2 transformed.

2) how was the cut-off (=2) for defining differential regulation chosen? Why no statistics was applied?

In a prior study, Martin et al. analyzed acetylation of one dozen histone sites by immunoblotting and showed that all sites, except H3K27ac, showed >2 -fold decrease in transcription (PMID: 33431884). A two-fold cut-off is also widely used in proteomic studies. Therefore, we chose this cut-off for our analyses. Following the reviewer's suggestion, we analyzed statistical significance for all sites that were quantified in at least 3 biological replicates. The Benjamini-Hochberg corrected p-values as well as log2 transformed fold changes were added to the supplemental table.

3) were different histone isoforms quantified separately or were they collapsed? This is another important point, in light of the fact that various histone isoforms (e.g. H3.1 vs H3.3 have been linked to chromatin in distinct transcriptional states and should be analyzed, possibly, individually)

In the data presented in Figure 3, sites occurring in different histone isoforms were collapsed. For each site, and for each replicate, the ratios of different histone isoforms were collapsed by calculating a median ratio per site, e.g. K16 at H2B Type-B, H2B Type-F, H2B Type-P were combined into H2B K16. The median of each replicate was then plotted in figure 3 as a circle and the bar indicates the mean across all replicates with a corresponding error bar.

Following reviewer's suggestion, we provide a separate table with all peptide forms of acetylated detected in histones, including isoform-specific peptides. In the figure, we feel that including all sites from individual isoforms makes the overwhelmingly busy to interpret, but if the reviewer recommends, we will present them individually for each isoform.

While acknowledging the biological importance of different histone isoforms, it should be noted neither of the prior studies that analyzed the link between histone acetylation and transcription distinguished acetylation in different isoforms, and there is no concrete evidence that transcription inhibition affects acetylation in an isoform-specific manner.

One last consideration is that -of the two alternative models of acetylation as cause of consequence of transcription, only the latter model has been (partially) addressed by the authors while no attempt to address the former option through a similar experimental design is present. Hence, with the current set of experiments/analyses (even upon the improvement of analysis of histone acetylation aforementioned), the manuscript seems to fit more to a "letter" format rather than to a "full article" one, as its information content and message are not comparable to those presented in the "articles" usually published by the journal. Since, apparently, the "Letter" is not an optional format for Nat Communications, this reviewer has major concern about the overall appropriateness of this submission.

We would like to emphasize that the objective of our work is to specifically test the recently proposed model that ongoing transcription is required for recruitment/activation of acetyltransferases and acetylation of histones (PMID: 33431884, PMID: 35273399, PMID: 35273400). The specific questions that we address are clearly outlined in the introduction section of the manuscript, and in the discussion, we transparently discuss that the question of causality has not been addressed in this work (or in the work that reported that histone acetylation is a consequence of ongoing transcription, PMID: 33431884, PMID: 35273399, PMID: 35273400).

"The model that ongoing transcription is required for the recruitment and/or activation of acetyltransferases and histone acetylation raises several fundamental questions. (1) Is transcription required for the acetylation of all histone sites or specific sites? (2) Does transcription specifically activate acetyltransferases for catalyzing histone acetylation? (3) What is the impact of transcription on the acetylation of non-histone proteins? The last question is pertinent here because the same acetyltransferases, such as GCN5/PCAF and CBP/p300, can catalyze acetylation both on histone and non-histone proteins 6,37-39."

"An important question for future studies is to clarify the role of histone and non-histone acetylation in transcription. This work was not designed to evaluate the functional roles of acetylation in transcription, and we present no claim, explicit or implied, to suggest a functional role of acetylation in gene regulation. Because multiple histone and non-histone proteins are acetylated in vivo, deciphering the exact role of acetylation in transcription regulation is a daunting task that will

require innovative approaches and community-wide efforts. Our results rule out a direct role of ongoing transcription in acetylation and thereby narrow down the number of hypotheses that need to be considered.”

While the question of causality is beyond the scope of this work, we would like to point out that our recent study has demonstrated that the catalytic inhibition of CBP/p300 impairs histone acetylation and transcription almost instantaneously (within 5 minutes) (PMID: 33765415, PMID: 37024579). Importantly, the transcription inhibition caused by CBP/p300 inhibition is rapidly restored by subsequent addition of lysine deacetylase inhibitors. To the best of our knowledge, this is the most direct demonstration of the importance of an acetyltransferase activity for the activation of hundreds of genes in mammalian cells, at such rapid kinetics. To us, these data imply (but do not demonstrate) a possible causal role of CBP/p300 activity in gene activation. However, this does not allow us to establish a causal role of histone acetylation in gene activation because CBP/p300 acetylates over one thousand sites in the cells. Therefore, it is exceedingly difficult to know whether acetylation of histone or non-histone proteins is required for CBP/p300-dependent gene activation. Because of this difficulty, it is exceedingly difficult to demonstrate a direct causal function of histone acetylation in gene activation, this is the reason why the cause versus consequence question is still debated 60 years after the discovery of histone acetylation.

As the reviewer notes, many journals, including Nature Communications, no longer make the distinction between different publication formats. We kindly request the reviewer to recognize this change in the journal's policy and instead consider the importance of the question being addressed in the work. In this context, the reviewer notes that “the analysis of histone acetylation in dependence of transcription, which is -as the authors clearly and correctly state in the Abstract- the most important open question to be addressed.” We fully agree on this view.

Here, we would like to draw reviewer’s attention to the fact that studies addressing the same question have been published in top-ranked journals, such as Nature Communications and Nature Genetics, in an article format, with accompanying expert views (PMID: 33431884, PMID: 35273399, PMID: 35273400). Rapid citation accumulation of these studies attests to the importance of the question being addressed, but doubts have been raised about these findings (PMID: 35768402). Therefore, a detailed analysis of acetylation is required to help resolve this important debate. By quantifying >20,000 acetylation sites, this work provides the most comprehensive analysis of changes in histone and non-histone acetylation after transcription inhibition. Since Nature Communications have published similar work recently, we sincerely believe that our findings deserve consideration at this journal.

The question of causality is a separate question for future work; we make no claim on this, and this is explicitly stated in the manuscript.

Reviewers' Comments:

Reviewer #1:

Remarks to the Author:

I find the data interesting and contradicting to previous publications, I am overall satisfied with the revised version.

However, I suggest reanalysis of ChIPseq data to be more informative.

Authors can divide genes to High and medium and low expressing based on the EU-seq data (from supplementary fig 1) in the control and show direct comparison by plotting K-means heatmap to show the correlation of H3K27ac with gene expression with control and upon transcription inhibition.

Reviewer #2:

Remarks to the Author:

Thanks for the clarification on the antibodies used, and I appreciate that concordant results were seen in similar experiments with a second primary antibody, and that this is encouraging. I also appreciate that the antibodies used are not new and have been used previously in multiple publications, but unfortunately this is not sufficient to determine the specificity of the antibody.

The use of clone D5E4 from CST however illustrates the challenges around specificity in the use of antibodies. This antibody shows comparable affinity with off target H3 peptide PTMs in a peptide array experiment (<http://www.histoneantibodies.com>). My concerns remain that without some method/model relevant validation of specificity the most that can be said from the data is that the affinity between the antibodies and reactive epitopes in mESCs is unchanged following incubation with inhibitors of transcription and is decreased following incubation with CBP/p300 inhibitors.

The question of how antibody specificity can be determined is extremely challenging. In the absence of any possible biological negative controls I suspect the next best thing would be an array based assessment of specificity against similar epitopes known to be expressed in the same tissue as the experiment, or selecting a monoclonal ab clone that has already been assessed in that way.

Reviewer #3:

Remarks to the Author:

The authors have now provided H3K27ac ChIP-seq data, as well as transcription data that supports their main conclusion drawn through proteomics. While this new study contradicts the conclusions drawn by two other studies (Martin et al., and Wang et al.), this nonetheless is an important study. The authors state that "However, our results contrast with some of the recent reports showing that histone acetylation is strongly reduced after transcription inhibition 34,35. The reason for this discrepancy is unclear."--I would like the authors to at least attempt to explain why they observed no change in acetylation by ChIP-seq, while the Wang et al study did

Reviewer #4:

Remarks to the Author:

Reviewer #4 has thoroughly evaluated the revised manuscript by Liebner et al., taking into account the authors' responses to the criticisms raised by all reviewers. Special attention was given to how the authors addressed the specific comments and criticisms outlined by this reviewer. Overall, it is concluded that the authors have satisfactorily addressed these criticisms, either through compelling explanations in the rebuttal letter or through amendments to the text,

including additional detailed information that helped clarify certain issues. Consequently, from this perspective, the revision addresses the points raised by reviewer #4.

However, when considering the manuscript as a whole, several points warrant highlighting:

1) The reviewer maintains reservations regarding the ChIP-seq validation experiment, suggesting that it should not solely focus on H3K27ac, a histone acetylation site known for its strong association with active cis-regulatory regions. Additional ChIP-seq experiments targeting other histone acetyl-Ks, such as H4K5/K8/12/16 or acetylations at H3K9/K14/K18/K23, which are more broadly distributed along gene bodies and other genomic regions but yet generally linked to open, actively transcribed regions, should be included. Given the significance and novelty of the concept being presented, and the high-quality journal they aim at publishing in, this reviewer suggests complementing IF and MS bulk data with genome-wide analysis of quantitative ChIP-seq data targeting additional histone Kac sites.

2) The relevance and adherence to the topic of the global acetyl-proteome profiling remain questionable to this reviewer. Despite the authors' justifications and comments, this aspect appears tangential to the core question of the causative link between transcription and histone acetylation. While acknowledging the depth and quality of the analysis, the reviewer feels that this part may be unnecessary for conveying the main message of the manuscript. Given the authors' emphasis on the significance of their study regarding the fundamental concept of establishing a causative relationship between transcription and acetylation, it is suggested that the entirety of the study be more consistently oriented towards the mechanisms underlying histone and chromatin acetylation. Other aspects may possess limited relevance within this specific context.

3) Throughout the rebuttal process, the authors have displayed reluctance to incorporate additional experiments or investigational changes based on relevant comments or criticisms from the reviewers. The reviewer observes that the authors' responses often lack consideration for their colleagues' viewpoints, reflecting a missed opportunity to strengthen the theoretical conceptual model by addressing such criticisms.

4) Considering that the authors have primarily engaged in a theoretical debate rather than executing new experiments or producing corroborating data, the reviewer suggests an additional assessment of the revised manuscript by a reviewer with strong theoretical expertise in the correlational/causal link between transcriptional regulation and histone acetylation. This additional review could provide another independent evaluation, as reviewer #4 feels insufficiently qualified to engage at this highly theoretical level of debate.

RESPONSE TO REVIEWERS' COMMENTS

Reviewer #1

I find the data interesting and contradicting to previous publications, I am overall satisfied with the revised version.

However, I suggest reanalysis of ChIPseq data to be more informative.

Authors can divide genes to High and medium and low expressing based on the EU-seq data (from supplementary fig 1) in the control and show direct comparison by plotting K-means heatmap to show the correlation of H3K27ac with gene expression with control and upon transcription inhibition.

We thank the reviewer for evaluating the revised manuscript. We are pleased to note that the reviewer is satisfied with the revised manuscript.

The correlation between gene expression and histone marks in basal (control) condition has been examined by many previous studies. Because H3K27ac shows no appreciable change after transcription inhibition, the correlation between gene expression level and H3K27ac in control and untreated cells remain virtually unchanged.

Given this result, it is a bit unclear to us what values should be used for K-means clustering. Nonetheless, as suggested, we divided genes according to their expression (high, medium, low) and analyzed H3K27ac in control and upon transcription inhibition and showed them in a heatmap (Extended Data Fig. 4).

If the reviewer suggests analyzing the data in a different way, it would be helpful if the reviewer could kindly clarify what exactly he/she expects us to do. If the reviewer could refer us to any prior studies conducting similar analyses, it would greatly assist us. We apologize for any inconvenience but aim to prevent any misunderstandings.

Reviewer #2:

Thanks for the clarification on the antibodies used, and I appreciate that concordant results were seen in similar experiments with a second primary antibody, and that this is encouraging. I also appreciate that the antibodies used are not new and have been used previously in multiple publications, but unfortunately this is not sufficient to determine the specificity of the antibody.

The use of clone D5E4 from CST however illustrates the challenges around specificity in the use of antibodies. This antibody shows comparable affinity with off target H3 peptide PTMs in a peptide array experiment (<http://www.histoneantibodies.com>). My concerns remain that without some method/model relevant validation of specificity the most that can be said from the data is that the affinity between the antibodies and reactive epitopes in mESCs is unchanged following incubation with inhibitors of transcription and is decreased following incubation with CBP/p300 inhibitors.

The question of how antibody specificity can be determined is extremely challenging. In the absence of any possible biological negative controls I suspect the next best thing would be an array based assessment of specificity against similar epitopes known to be expressed in the same tissue as the

experiment, or selecting a monoclonal ab clone that has already been assessed in that way.

We are happy to note that the reviewer appreciates the inclusion of data from an additional antibody.

Concerning the suggestions to further test antibody specificity, we would like to clarify that the specificity of both antibodies has been tested in the way the reviewer suggests. The antibody from CST (clone D5E4) has been validated most rigorously by mutating H3K27 *in vivo* (PMID: 35668298). In cells mutated of H3K27, the H3K27ac antibody shows no signal in immunoblotting and in ChIP-seq (PMID: 35668298), only a background detectable signal in ChIP-seq from 'Phantom peaks' (PMID: 26117547).

For the second antibody, (Rabbit monoclonal, Abcam), the specificity has been evaluated using an array-based method (as the reviewer requests) and the assessment data are publicly available here.

<https://www.abcam.com/en-dk/products/primary-antibodies/histone-h3-acetyl-k27-antibody-ep16602-chip-grade-ab177178#overlay=images&application=peparr>

Based on these data, we believe that the specificity of antibodies, which are widely used in prior publications, have already been rigorously demonstrated, and therefore, we see no reason to repeat these characterizations.

Reviewer #3 (Remarks to the Author):

The authors have now provided H3K27ac ChIP-seq data, as well as transcription data that supports their main conclusion drawn through proteomics. While this new study contradicts the conclusions drawn by two other studies (Martin et al., and Wang et al.), this nonetheless is an important study. The authors state that "However, our results contrast with some of the recent reports showing that

histone acetylation is strongly reduced after transcription inhibition 34,35. The reason for this discrepancy is unclear."--I would like the authors to at least attempt to explain why they observed no change in acetylation by ChIP-seq, while the Wang et al study did

We appreciate the reviewer's effort in evaluating the revised manuscript and appreciate his/her positive assessment.

Several studies have explored the impact of transcription inhibition on histone acetylation, yielding conflicting results. Therefore, we feel that it is important to discuss the findings in the context of overall literature, rather than singling out any specific work. We must acknowledge that we simply do not know why the findings of Wang et al. differ from our results and of several other previously published papers. Without understanding the likely reasons for the discrepancy, we are hesitant to engage in speculative discussions about a specific work. Instead, we have addressed the disparity in our findings compared to prior literature within a broader framework.

"Our findings are consistent with early studies showing that histone acetylation is not affected by transcription inhibition^{13,14,50,51}. However, our results contrast with some of the recent reports showing that histone acetylation is strongly reduced after transcription inhibition^{34,35}. The reason for this discrepancy is unclear. One of the studies showed that H3K27ac and H3K27me3 were similarly strongly decreased within 1 hour of transcription inhibition³⁵. This is perplexing given that the half-lives of H3K27ac and H3K27me3 are very different in mammalian cells^{52,53}. The rapid reduction of H3K27me3 in EZH2-bound peak regions was rationalized by transcription-dependent recruitment of EZH2^{35,54}. However, H3K27me3 decreases very slowly after the chemical or genetic ablation of EZH2 function, and the rate of H3K27me3 reduction roughly equals the time of the cell division cycle, indicating that the decrease in H3K27me3 after EZH2 activity ablation occurs by replication-coupled histone dilution rather than active demethylation^{55,56}."

Reviewer #4 (Remarks to the Author):

Reviewer #4 has thoroughly evaluated the revised manuscript by Liebner et al., taking into account the authors' responses to the criticisms raised by all reviewers. Special attention was given to how the authors addressed the specific comments and criticisms outlined by this reviewer. Overall, it is concluded that the authors have satisfactorily addressed these criticisms, either through compelling explanations in the rebuttal letter or through amendments to the text, including additional detailed information that helped clarify certain issues. Consequently, from this perspective, the revision addresses the points raised by reviewer #4.

We thank the reviewer for evaluating the revised manuscript. We appreciate the reviewer's acknowledgment that the original comments were adequately addressed.

However, when considering the manuscript as a whole, several points warrant highlighting:

1) The reviewer maintains reservations regarding the ChIP-seq validation experiment, suggesting that it should not solely focus on H3K27ac, a histone acetylation site known for its strong association with active cis-regulatory regions. Additional ChIP-seq experiments targeting other histone acetyl-Ks, such as H4K5/K8/12/16 or acetylations at H3K9/K14/K18/K23, which are more broadly distributed along gene bodies and other genomic regions but yet generally linked to open, actively transcribed regions, should be included. Given the significance and novelty of the concept being presented, and the high-

quality journal they aim at publishing in, this reviewer suggests complementing IF and MS bulk data with genome-wide analysis of quantitative ChIP-seq data targeting additional histone Kac sites.

In his/her original remarks, the reviewer did not raise any comment about ChIP-seq experiments. Instead, the reviewer mainly raised concerns that the mass spectrometry approach used in our work was not suitable for the quantification of histone sites. We thoroughly address this concern by clarifying this misunderstanding.

Our main conclusion is that the **global level** of histone and non-histone proteins remain virtually unchanged after transcription inhibition, and we are confident that our data robustly support this conclusion. Whether transcription inhibition locus-specifically impact histone sites other than H3K27ac remains to be investigated in future efforts. We are unaware of evidence showing that different histone sites are impacted differently by transcription inhibition. Nonetheless, to acknowledge this possibility in the discussion.

“Additionally, our analysis of protein acetylation, including histones, is conducted in bulk. It is hypothetically possible that transcription inhibition may decrease acetylation at specific histone sites or combinations only, or that the loss of acetylation in some genomic regions is compensated by an increase in acetylation in others.”

2) The relevance and adherence to the topic of the global acetyl-proteome profiling remain questionable to this reviewer. Despite the authors' justifications and comments, this aspect appears tangential to the core question of the causative link between transcription and histone acetylation. While acknowledging the depth and quality of the analysis, the reviewer feels that this part may be unnecessary for conveying the main message of the manuscript. Given the authors' emphasis on the significance of their study regarding the fundamental concept of establishing a causative relationship between transcription and acetylation, it is suggested that the entirety of the study be more consistently oriented towards the mechanisms underlying histone and chromatin acetylation. Other aspects may possess limited relevance within this specific context.

We acknowledge reviewer's view that some readers may be interested only in effect of transcription inhibition on histone sites. We kindly request the reviewer to also recognize that there are others who may be interested in knowing whether the effect of transcription inhibitors on acetylation is restricted to histones or also affect non-histone proteins. Therefore, we feel that removing non-histone data, and only focusing on histone acetylation would provide an incomplete picture of the impact of transcription inhibition on global acetylation. In our previous responses, we provided the rationale for studying both histone and non-histone protein acetylation. We kindly request the reviewer to refer to our previous rebuttal (pages 14-15) for further clarification.

3) Throughout the rebuttal process, the authors have displayed reluctance to incorporate additional experiments or investigational changes based on relevant comments or criticisms from the reviewers. The reviewer observes that the authors' responses often lack consideration for their colleagues' viewpoints, reflecting a missed opportunity to strengthen the theoretical conceptual model by addressing such criticisms.

We apologize if our responses lacked consideration of reviewers/colleagues' viewpoints. We addressed all the points raised to the best of our ability, and as far as we can see, all four reviewers have acknowledged that their comments have been adequately addressed.

4) Considering that the authors have primarily engaged in a theoretical debate rather than executing new experiments or producing corroborating data, the reviewer suggests an additional assessment of the revised manuscript by a reviewer with strong theoretical expertise in the correlational/causal link between transcriptional regulation and histone acetylation. This additional review could provide another independent evaluation, as reviewer #4 feels insufficiently qualified to engage at this highly theoretical level of debate.

This appears a misunderstanding, perhaps arising from unclarity in our responses. We would like to clarify that we have no intention to enter a debate with reviewers. We only intend to address their concerns objectively and comprehensively. We apologize if this was not clear. We leave it to the editor's discretion to determine whether input from additional reviewer(s) is required.

Reviewers' Comments:

Reviewer #1:

Remarks to the Author:

In this revision, As requested, the authors have divided the heatmap based on the expression level. I was expecting to see a clustering analysis, which might reveal some loci with reduced levels of acetylation compared to the majority of the loci which could be due to the transcriptional status of the loci.

However, further analysis of H3K27ac ChIPseq data based on expression level clearly shows that H3K27ac is not reduced at transcription sites upon inhibition of transcription. The data convincingly shows that the acetylation level does not change in all types of genes, irrespective of the level of transcription.

Overall, this work provides strong mass spec and ChIPseq-based quantification data to support their conclusion.